

# Intra-specific variation and allometry of the skull of Late Cretaceous side-necked turtle *Bauruemys elegans* (Pleurodira, Podocnemididae) and how to deal with morphometric data in fossil vertebrates

Thiago F. Mariani[1,2] and Pedro S.R. Romano[1]

[1] Deparatamento de Biologia Animal, Universidade Federal de Viçosa, Viçosa, MG, Brazil
[2] Current affiliation: Departamento de Geologia e Paleontologia, Museu Nacional, Universidade Federal do Rio de Janeiro, Rio de Janeiro, RJ, Brazil

Corresponding author
Thiago F. Mariani,
tmariani.bio@gmail.com

## ABSTRACT

**Background.** Previous quantitative studies on *Bauruemys elegans* (Suárez, 1969) shell variation, as well as the taphonomic interpretation of its type locality, have suggested that all specimens collected in this locality may have belonged to the same population. We rely on this hypothesis in a morphometric study of the skull. Also, we tentatively assessed the eating preference habits differentiation that might be explained as due to ontogenetic changes.

**Methods.** We carried out an ANOVA testing 29 linear measurements from 21 skulls of *B. elegans* taken by using a caliper and through images, using the ImageJ software. First, a Principal Components Analysis (PCA) was performed with 27 measurements (excluding total length and width characters; =raw data) in order to visualize the scatter plots based on the form variance only. Then, a second PCA was carried out using ratios of length and width of each original measurement to assess shape variation among individuals. Finally, original measurements were log-transformed to describe allometries over ontogeny.

**Results.** No statistical differences were found between caliper and ImageJ measurements. The first three PCs of the PCA with raw data comprised 70.2% of the variance. PC1 was related to size variation and all others related to shape variation. Two specimens plotted outside the 95% ellipse in PC1~PC2 axes. The first three PCs of the PCA with ratios comprised 64% of the variance. When considering PC1~PC2, all specimens plotted inside the 95% ellipse. In allometric analysis, five measurements were positively allometric, 19 were negatively allometric and three represented enantiometric allometry. Many bones of the posterior and the lateral emarginations lengthen due to increasing size, while jugal and the quadratojugal decrease in width.

**Discussion.** ImageJ is useful in replacing caliper since there was no statistical differences. Yet iterative imputation is more appropriate to deal with missing data in PCA. Some specimens show small differences in form and shape. Form differences were interpreted as occuring due to ontogeny, whereas shape differences are related to feeding changes during growth. Moreover, all outlier specimens are crushed and/or distorted, thus the form/shape differences may be partially due to taphonomy. The allometric lengthening of the parietal, quadrate, squamosal, maxilla, associated with the

narrowing of jugal and quadratojugal may be related to changes in feeding habit between different stages of development. This change in shape might represent a progressive skull stretching and enlargement of posterior and lateral emargination during ontogeny, and consequently, the increment of the feeding-apparatus musculature. Smaller individuals may have fed on softer diet, whereas larger ones probably have had a harder diet, as seen in some living species of *Podocnemis*. We conclude that the skull variation might be related to differences in feeding habits over ontogeny in *B. elegans*.

## INTRODUCTION

### Principal Component Analysis and fossil sampling bias

Paleontological data are intrinsically scarce (*Strauss, Atanassov & Oliveira, 2003*; *Hammer & Harper, 2006*), leading to incomplete data sampling. This limitation impacts several approaches in paleontological studies, especially inter-specific variation analyses. Although there are some methods proposed to deal with missing entries in fossil quantitative datasets (e.g., *Norell & Wheeler, 2003*; *Strauss, Atanassov & Oliveira, 2003*), sometimes the study relies on an exploratory evaluation of general structure and Principal Component Analysis (PCA) is commonly used for this purpose.

PCA is a method to ordinate multivariate data. Its aim is to identify the variables that account for the majority of the variance within a multivariate matrix, by means of linear combinations of all variables, which are converted into components that are independent of each other (*Strauss, Atanassov & Oliveira, 2003*; *Hammer, Harper & Ryan, 2001*). Hence, PCA summarizes a large amount of the variance contained in the data (*Krzanowski, 1979*; *Hammer, Harper & Ryan, 2001*). It thus reduces a multidimensional space into fewer components which retain the majority of the variance of a given sample (*Jolicoeur & Mosimann, 1960*; *Peres-Neto, Jackson & Somers, 2003*), and is therefore an useful tool for exploring large, complex data sets, being largely applied to both extant and turtles (e.g., *Jolicoeur & Mosimann, 1960*; *Claude et al., 2004*; *Depecker et al., 2005*; *Depecker et al., 2006*; *Werneburg et al., 2014*; *Ferreira et al., 2015*).

### Case-study

#### Skull variation

The skull is one of the most variable structures in vertebrates because it concentrates several sensory organs, the brain, and the beginning of the respiratory and digestory systems, including chewing muscles (*Smith, 1993*). Consequently, the skull is the body part with more phenotypes used in vertebrate cladistic analysis (*Rieppel, 1993*), as seen in turtles, in which most cladistic analysis rely mainly on cranial characters (e.g., *Gaffney, Meylan & Wyss, 1991*; *De La Fuente, 2003*; *Gaffney, Tong & Meylan, 2006*; *Gaffney et al., 2011*; *Joyce, 2007*; *Joyce & Lyson, 2010*; *Sterli et al., 2010*; *Sterli & De La Fuente, 2011*; *Anquetin, 2012*; *Rabi et al., 2013*; *Romano et al., 2014*; *Ferreira et al., 2015*; *Romano, 2016*). Despite

that, most skull materials found in paleontological record of turtles are rare and/or damaged due to the fossilization process bias, not allowing intraspecific comparisons or ontogenetic inferences on most fossil turtle species known. Some exceptions are found in *Sánchez-Villagra & Winkler (2006)* and *Ferreira et al. (2015)*, who performed interspecific comparisons among fossil turtle taxa using skull material.

### *Bauruemys* taxonomy

*Bauruemys elegans* (Suárez, 1969) is a Late Cretaceous freshwater side-necked turtle found at the Pirapozinho site (*Suárez, 2002*), in western São Paulo state. This species was originally described as *Podocnemis* in three different communications by *Suárez (1969a)*, *Suárez (1969b)* and *Suárez (1969c)* and such recognition was based on the overall similarities of the skull and shell to this living genus, a common practice that time. Other South American Cretaceous side-necked turtles were initially identified as *Podocnemis* as well, such as the *nomina dubia* "*Roxochelys*" *harrisi* (Pacheco, 1913) and "*Bauruemys*" *brasiliensis* (Staeche, 1937) and the *incertae sedis* "*Podocnemis*" *argentinensis* (Cattoi & Freiberg, 1958) (see *Romano et al., 2013* for a revision on Bauru Group species). *Kischlat (1994)* was the first to point out that all *Podocnemis* reported from the Cretaceous were doubtful and proposed a new genus, *Bauruemys*, to include *B. elegans* and, tentatively, *B. brasiliensis.* His conclusion was based on similarities of the plastron of both species, besides the analysis of cranial features in *B. elegans.* More recently, *Romano et al. (2013)* confirmed the recognition of *B. brasiliensis* as *Bauruemys,* but considering this species as *nomem dubium. Kischlat (1994)* and *Kischlat, Barbarena & Timm (1994)* also pointed out that *B. elegans* could belong to Podocnemididae, but they did not test their hypothesis. *Romano & Azevedo (2006)* were the first to carry out a cladistic analysis to access the phylogenetic position of *Bauruemys*, placing it as a stem-Podocnemididae, i.e., the sister group of all other Podocnemididae, which was consistently confirmed by subsequent analyses with more podocnemidid species included as terminals (*França & Langer, 2005*; *Gaffney et al., 2011*; *Oliveira, 2011*; *Cadena, Bloch & Jaramillo, 2012*).

### Geological settings and taphonomic context of the Tartaruguito site

The Pirapozinho site, long ago known as "Tartaruguito" and formally assigned as such by *Romano & Azevedo (2007)* and *Gaffney et al. (2011)*, is an Upper Cretaceous outcrop from the Presidente Prudente Formation, Bauru Basin (Geology *sensu Fernandes & Coimbra, 2000*). It is located in Pirapozinho municipality, São Paulo State, Brazil (Fig. 1). The "Tartaruguito" name, which means "turtle in rock" (*tartaruga*, from Portuguese, turtle; *ito*, from Latin, rock. More precisely, "ito" is the Portuguese suffix correspondent to "ite," the suffix used in geology to forms the names of rocks and minerals.), reflects the great amount of turtle specimens found at that place. It is comparable to other rich fossil turtle localities, such as (1) the recently discovered Middle Jurassic Qigu Formation of the Turpan Basin in China (*Wings et al., 2012*; *Rabi et al., 2013*); (2) the Late Cretaceous (Maastrichtian) Hell Creek Formation ('Turtle Graveyard') in Slope County, North Dakota, USA (*Lyson & Joyce, 2009*); (3) the Middle-Upper Paleocene Cerrejón Formation in Colombia (*Jaramillo et al., 2007*; *Cadena, Bloch & Jaramillo, 2010*; *Cadena, Bloch & Jaramillo, 2012*; *Cadena et al.,*
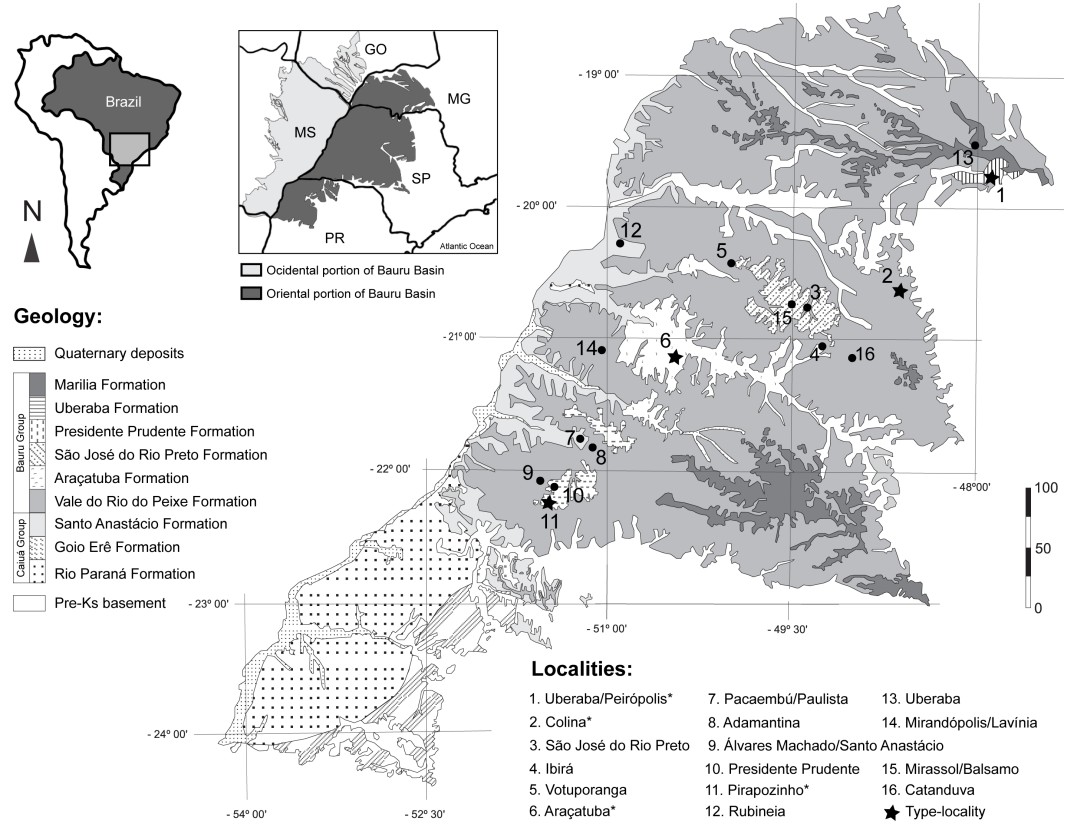

**Figure 1** **Fossil turtle localities in Bauru Basin.** Lithostratigraphical map of the oriental part of the Bauru Basin showing the fossil turtle localities (municipalities). Turtle species are: 1. *Cambaremys langertoni* (*incertae sedis*), *Pricemys caieira*, *Peiropemys mezzalirai* and Testudines indet. (*Oliveira & Romano, 2007*; *Romano et al., 2009*; *Gaffney et al., 2011*; *Menegazzo, Bertini & Mazini, 2015*); 2. *Roxochelys harrisi* (*nomem dubium*; *Oliveira & Romano, 2007*; *Romano et al., 2009*; *Menegazzo, Bertini & Mazini, 2015*; 3. *Bauruemys brasiliensis* (*nomem dubium*) and Testudines indet. (*Oliveira & Romano, 2007*; *Menegazzo, 2009*; *Romano et al., 2009*; *Menegazzo, Bertini & Mazini, 2015*; 4. Testudines indet. (*Menegazzo, 2009*; *Romano et al., 2009*); 5. Testudines indet. (*Menegazzo, 2009*; *Romano et al., 2009*); 6. *B. brasiliensis* and *Roxochelys wanderleyi* (*Oliveira & Romano, 2007*; *Romano et al., 2009*) 7. Testudines indet. (*Menegazzo, 2009*; *Romano et al., 2009*); 8. Testudines indet. (*Menegazzo, 2009*; *Romano et al., 2009*); 9. Podocnemididae indet. and Testudines indet. (*Menegazzo, 2009*; *Romano et al., 2009*; *Kischlat, 2015*); 10. *Roxochelys* sp., *R. wanderleyi* and Testudines indet. (*Menegazzo, 2009*; *Romano et al., 2009*; *Romano et al., 2013*; *Menegazzo, Bertini & Mazini, 2015*; 11. *B. elegans* and *R. wanderleyi* (*Oliveira & Romano, 2007*; *Romano et al., 2009*; *Menegazzo, Bertini & Mazini, 2015*; *Suárez, 1969b*); 12. Podocnemidinura indet. (*Menegazzo, Bertini & Mazini, 2015*); 13. Podocnemidoiae indet. and Testudines indet. (*Menegazzo, 2009*; *Hermanson, Ferreira & Langer, 2016*); 14. *R. wanderleyi*, *B. brasiliensis* (*nomem dubium*) and Testudines indet. (*Menegazzo, 2009*); 15. Testudines indet. (*Menegazzo, 2009*); 16. Testudines indet. (*Menegazzo, 2009*). Abbreviations: GO, Goiás State; MG, Minas Gerais State; MS, Mato Grosso do Sul State; PR, Paraná State; SP, São Paulo State. Scale bar in Km. Map modified from *Menegazzo, 2009* and *Romano et al. (2009)*; geology following *Fernandes (2004)*; taxonomy status of species following *Romano et al. (2013)*.

*2012*); and (4) the Upper Miocene Urumaco Formation ('Capa de tortugas') in Venezuela (*Aguilera, 2004*; *Sánchez-Villagra & Aguilera, 2006*; *Sánchez-Villagra & Winkler, 2006*; *Riff et al., 2010*; *De La Fuente, Sterli & Maniel, 2014*). The two latter localities are near-shore marine coastal deposits with influence of freshwater rivers (*Jaramillo et al., 2007*; *Gaffney et al., 2008*), whereas the two former and the Tartaruguito site correspond to sediments

that had been deposited in a riverine system with seasonal droughts in which turtles gathered in retreating, ephemeral water pools and died when habitat dried up completely (*Soares et al., 1980*; *Fulfaro & Perinotto, 1996*; *Fernandes & Coimbra, 2000*; *Henriques et al., 2002*; *Henriques et al., 2005*; *Suárez, 2002*; *Murphy, Hoganson & Johnson, 2003*; *Bertini et al., 2006*; *Henriques, 2006*; *Wings et al., 2012*). The Tartaruguito is also the type-locality of the peirosaurid crocodile *Pepesuchus deiseae* Campos, Oliveira, Figueiredo, Riff, Azevedo, Carvalho & Kellner, 2011 (*Campos et al., 2011*).

The general lithology of the Tartaruguito site is composed of cyclic alternations of sandstones and mudstones deposited in a meandering fluvial system with crevasse splays (*Fernandes & Coimbra, 2000*; *Henriques et al., 2005*; *Bertini et al., 2006*). Many articulated and complete fossils are found in these sequences, which indicate seasonal low energy floods (mudstones) followed by droughts (sandstones) in the region during the Late Cretaceous (*Henriques et al., 2002*; *Henriques et al., 2005*; *Henriques, 2006*). Because only medium- to large-sized fossil specimens are found at the locality, it is assumed that the Tartaruguito site was a foraging area for turtles (D Henriques & L Carvalho, pers. comm., 2016). Thus, the fossil assemblage probably represents several episodes of floods and droughts. The flood periods might have allowed foraging areas expansion for turtles and crocodiles, while during the dry seasons turtles gathered on the remnants of water pools and some died when pools dried up completely (*Henriques et al., 2002*; *Henriques et al., 2005*; *Henriques, 2006*).

That being said, we consider that all turtle specimens found at the Tartaruguito site might correspond to subadults to adult ages, and it is reasonable to assume that all *B. elegans* individuals collected in the Tartaruguito site might have belonged to a single population (agreeing with *Henriques et al., 2002*; *Henriques et al., 2005*; *Henriques, 2006*; *Romano & Azevedo, 2007*). Indeed, as suggested by *Romano & Azevedo (2007)*, this single population would consist on different generations of turtles' corpses grouped in the same locality. One might consider that size differences might be due to sexual dimorphism (R Hirayama & S Thomson, pers. comm., 2015), in which the females would be larger and have more posteriorly extended carapaces than the males. However, sexual dimorphism on podocnemidid turtles can be assessed only on shell shape and our data is based mostly on isolated skulls (see Material and Methods). As a consequence, although it is possible that sexual dimorphism may affect measurements captured in this study, we did not consider it, given the lack of evidence to assume such outcome. Also, to our knowledge, skull shape differences related to sexual dimorphism has never been described to podocnemidid turtles yet. Moreover, *Romano & Azevedo (2007)* were not able to reject the single population hypothesis using shell measurements (from both plastron and carapace) in a morphometric approach neither to describe sexual dimorphism in the data, concluding that the differences were due to ontogenetic variation. Therefore, we highlight that we are assuming the population definition of *Futuyma (1993)*, as taken on by *Romano & Azevedo (2007)*, that a population is a conjunct of semaforonts temporally connected, i.e., a sequence of individuals from different generations, and limited in a restricted space; in this case, the Tartaruguito site. By assuming this, we explicitly follow *Hennig*'s (*1966*) semaphoront concept, on which a species is modifiable (i.e., not strictly typological) and represented by a sequence of generations.

## Objectives

Efforts to study fossil materials may be hampered by difficulty in accessing foreign collections. It can narrow and even preclude their studies. In addition, given the missing data problem inherent to fossil record, the way one treats the missing entries in morphometric studies can affect the results and conclusions. Regarding the use of caliper or ImageJ in taking measurements, here we tested both approaches by taking linear measurements for morphometric studies based on photographs (e.g., *Bailey, 2004*) and also evaluated how different approaches designed to deal with missing data can impact results of exploratory statistical procedures and data interpretation by comparing two different substitution algorithms of missing entries. These procedures are exemplified using a real paleontological data set and with paleobiological inferences. Considering the case-study, we explored the variation in skulls among individuals of *Bauruemys elegans* from different ages and generations. Also, we describe the differences in skull morphology along the ontogeny of the species and discuss the probable consequences of such variation to the diet preferences changes along the growth.

## MATERIAL AND METHODS

### Sample and characters

Twenty-one skulls of *Bauruemys elegans* were examined in this study, including the type series plus nineteen topotypes: AMNH-7888, LPRP0200, LPRP0369, LPRP0370, MCT 1492-R (holotype), MCT 1753-R (paratype), MCZ 4123, MN 4322-V, MN 4324-V, MN 6750-V, MN 6783-V, MN 6786-V, MN 6787-V, MN 6808-V, MN 7017-V, MN 7071-V, MZSP-PV29, MZSP-PV30, MZSP-PV32, MZSP-PV34, and MZSP-PV35. We established 39 landmarks (Fig. 2) that decompose the overall shape of the skull in order to take measurements between two landmarks. Since most of the specimens have deformation and breakage, we could not perform a geometric morphometric analysis using the landmarks because the taphonomical bias would incorporate error to the analysis of form and shape. Thus, we used the landmarks to set up 29 traditional morphometric characters that correspond to a linear measurement between two landmarks (all characters are described in Table 1). Also, the use of landmarks to set up the measurements is useful to maintain the same anatomic references for all characters in each specimen, since the landmarks enable a better description of morphological variation and establishment of linear measurements, as performed by *Romano & Azevedo (2007)* with shell morphometric characters. All measurements were taken on the same side of the skull (right side) unless the characters could not be measured due to deformation or breakage. We are aware that deeper structures (*z*-axis) can influence the straight line between two landmarks in 2D images and used ImageJ version 1.47 (*Rasband, 1997*) to take the measurements after comparing its accuracy with the caliper (*Mariani & Romano, 2014*). This procedure was necessary because we obtained photos of skulls in dorsal, ventral, and lateral views housed in foreign collections and did not collect the described measurements (see Table 1) using caliper in such specimens because they were analyzed prior to this study. The error test between measurements taken using caliper and ImageJ using part of the sample are

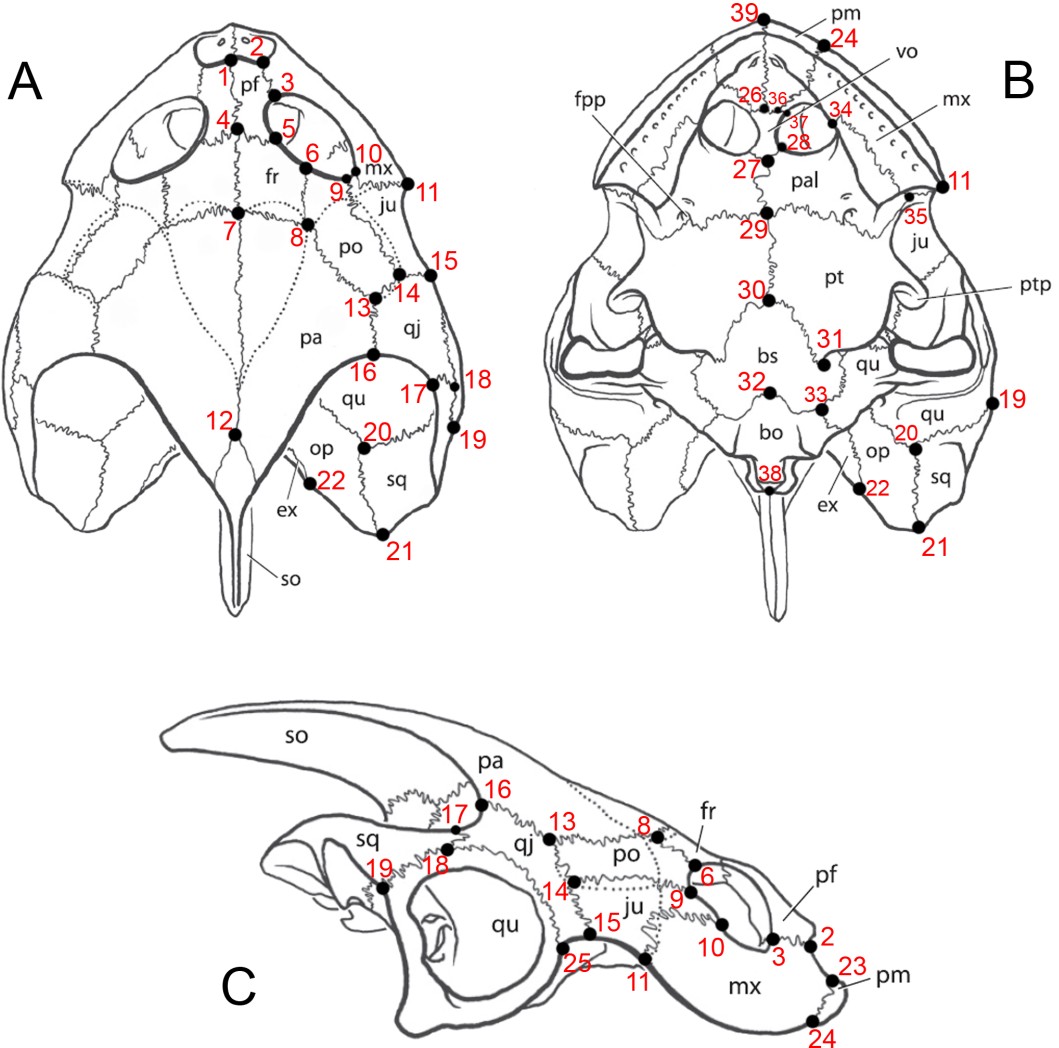

**Figure 2   Image of landmarks used as references for taking measurements.** Skull of *Bauruemys elegans* in (A) dorsal, (B) ventral and (C) right lateral views showing the anatomical nomenclature and the 39 landmarks used for morphometrics analysis. All measurements were taken between two landmarks (see Table 2 for vectors description). Abbreviations: bo, basioccipital; bs, basisphenoid; ex, exoccipital; fpp, foramen palatinum posterius; fr, frontal; ju, jugal; mx, maxilla; op, opisthotic; pa, parietal; pal, palatine; pf, prefrontal; pm, premaxilla; po, postorbital; pt, pterygoid; ptp, processus trochlearis pterygoidei; qj, quadratojugal; qu, quadrate; sq, squamosal; so, supraoccipital; vo, vomer. Skull lineation from *Gaffney et al.* (*2011*, p.72).

described below. We followed the bone nomenclature of *Parsons & Williams (1961)* and extended by *Gaffney (1972)* and *Gaffney (1979)* (see all abbreviations after Conclusion topic).

## Statistical analyses
### Preliminary analysis: caliper vs. ImageJ
Before carrying out others statistical analyses, we compared the same characters data set (Data S1) of a sub-sample by using two different approaches (=treatments): measurements

**Table 1 ANOVA results for ImageJ and caliper comparisons.** Parameters calculated for each treatment of the ANOVA. Columns 2, 3, and 4 are relative to the caliper (cal), columns 5, 6 and 7 are relative to the ImageJ (ImJ). The last column indicates the *F* values for each character.

| Char. | N (Cal) | Mean (Cal) | σ (Cal) | N (ImJ) | Mean (ImJ) | σ (ImJ) | *F* value |
|---|---|---|---|---|---|---|---|
| TLS | 8 | 63.72 | 10.87 | 8 | 62.26 | 11.36 | 0.069 |
| TWS | 9 | 60.42 | 9.45 | 8 | 64.83 | 13.58 | 0.617 |
| LPF | 9 | 9.78 | 1.26 | 9 | 8.05 | 1.80 | 5.617[*] |
| WPF | 10 | 6.70 | 1.90 | 10 | 7.55 | 1.83 | 1.04 |
| LFR | 10 | 12.19 | 1.74 | 10 | 11.79 | 2.02 | 0.233 |
| WFR | 10 | 9.64 | 1.63 | 10 | 10.12 | 1.82 | 0.383 |
| LPA | 7 | 25.54 | 4.71 | 7 | 27.35 | 4.83 | 0.504 |
| WPA | 6 | 21.78 | 2.79 | 6 | 22.54 | 3.16 | 0.195 |
| SMX | 9 | 46.46 | 7.12 | 9 | 47.66 | 8.62 | 0.104 |
| LVO | 6 | 5.95 | 1.71 | 7 | 6.59 | 1.31 | 0.596 |
| WVO | 6 | 3.11 | 0.78 | 7 | 3.68 | 0.52 | 1.874 |
| WCO | 5 | 7.53 | 1.31 | 6 | 6.45 | 1.15 | 2.107 |
| LPAL | 7 | 8.26 | 1.25 | 8 | 7.21 | 2.81 | 0.828 |
| WPAL | 7 | 16.90 | 1.91 | 7 | 17.12 | 2.23 | 0.038 |
| LPT | 11 | 11.54 | 2.06 | 12 | 11.69 | 2.75 | 0.228 |
| LBS | 12 | 12.43 | 1.30 | 12 | 12.88 | 1.64 | 0.563 |
| WBS | 11 | 15.58 | 2.32 | 11 | 15.57 | 2.40 | <0.001 |
| Lbo | 7 | 13.00 | 1.84 | 7 | 13.84 | 1.85 | 0.726 |
| LMS | 10 | 24.28 | 4.20 | 9 | 19.22 | 4.15 | 6.937[*] |
| WMX | 10 | 10.44 | 2.16 | 9 | 10.18 | 2.26 | 0.065 |
| LJU | 9 | 15.75 | 3.81 | 7 | 13.39 | 2.92 | 1.847 |
| WJU | 3 | 8.31 | 1.20 | 2 | 9.83 | –[**] | 2.709 |
| LQJ | 4 | 12.84 | 1.48 | 2 | 11.96 | –[**] | 0.366 |
| WQJ | 6 | 16.21 | 4.02 | 3 | 19.65 | 1.72 | 1.921 |
| LQU | 11 | 17.71 | 3.43 | 8 | 21.19 | 3.88 | 4.253 |
| LPO | 9 | 16.57 | 3.30 | 9 | 16.89 | 4.11 | 0.35 |
| WPO | 9 | 5.47 | 1.77 | 8 | 5.44 | 1.73 | 0.002 |
| WOP | 6 | 11.97 | 2.52 | 5 | 10.98 | 3.89 | 0.260 |
| LSQ | 5 | 10.63 | 3.28 | 4 | 12.26 | 3.86 | 0.467 |

**Notes.**

Measurements abbreviations:: TLS, total length of the skull; TWS, total width of the skull; LPF, length of prefrontal; WPF, width of prefrontal; LFR, length of frontal; WFR, width of frontal; LPA, length of parietal; WPA, width of parietal; SMX, stretch of maxilla; LVO, length of vomer; WVO, width of vomer; WCO, width of choannal; LPAL, length of palatine; WPAL, width of palatine; LPT, length of pterygoid; LBS, length of basisphenoid; WBS, width of basisphenoid; LBO, length of basisoccipital; LMX, length of maxilla; WMX, width of maxilla; LJU, length of jugal; WJU, width of jugal; LQJ, length of quadratojugal; WQJ, width of quadratojugal; LQU, length of quadrate; LPO, length of postorbital; WPO, width of postorbital; WOP, width of opisthotic; LSQ, length of squamosal; Cal, caliper treatment; ImJ, ImageJ treatment.

[*]significant statistically differences.

[**]values not calculated.

taken using caliper and measurements taken using photographs via ImageJ. This comparison was necessary in order to evaluate whether or not the two measurements methods are significantly different. Then, we performed an One-way Analysis of Variance (ANOVA) comparing the 29 measurements in 12 specimens (LPRP0200, LPRP0369, LPRP0370, MN4322-V MN4324-V, MN6750-V, MN6783-V, MN6786-V, MN6787-V,
MN6808-V, MN7017-V, and MN7071-V). Two groups of variables were established: measurements taken directly from specimens using caliper (preliminary data set 1) and the same characters taken from photographs of the same specimens using ImageJ (preliminary data set 2). All characters taken using photographs/ImageJ that did not show significant differences to their correspondents taken by caliper were used on the subsequent statistical analyses of form and shape differences among the sample of *Bauruemys elegans*. By doing that, the sample was increased without including error and incomparable characters (i.e., by using different measurement techniques).

We found most of the measurements do not differ statistically ($p > 0.05$) between the two treatments (caliper and ImageJ; Table 1). However, one measurement, length of maxilla (LMX), had statistical difference ($p = 0.017$) between the treatments, because the maxilla is a curved structure and thus the landmarks are in different positions (LM 24 is deeper and farther from the camera in relation to LM11) in relation to the plane the picture was taken. Given that no statistical differences were found in almost all characters, ImageJ could be an economic and time-saving tool for morphometric analyses from photographs (2D), and could be applied by scientists at distant institutions.

The study *in situ* of the material is preferable, although pictures are an economic alternative in where cases one is not able to handle the material. We must aware that one have to choose one of the two treatments to construct a morphometric matrix, otherwise it will be composed of values obtained by two diffent methods.

### Univariate, multivariate and allometric analyses

Three analyses using the complete sample were carried out: (1) a descriptive statistics (mean, standard deviation, median, variance, maximum and minimum values) of all characters (Data S2), (2) an allometric analysis of length and width characters correlating them to total length and width measurements (Data S3), and (3) a multivariate non-parametric exploratory statistics via Principal Component Analysis (PCA). The latter was divided into two different PCAs: (3.1) using 27 characters from the raw data matrix (total length and width characters were excluded in this analysis; Data S4—because PCA is sensitive to large variations in the original measurements), and (3.2) using 27 characters that correspond to the proportions of each character from the raw data (i.e., original measurements) represented by its length or width characters divided by each individual total length or width (e.g., the length of MCZ4312 postorbital divided by the total length of the skull of this specimen; see complete data in Data S5). All statistical analyses were performed using the software PAST version 3.05 (*Hammer, Harper & Ryan, 2001*).

In the allometric analysis (analysis 2, Data S3), all characters were previously log-transformed and a linear regression was carried out separately for length and width characters, using the least square fitting approach for residuals. We established the allometries by considering the regression's slope, i.e., the coefficient *a*, as following: positive allometry ($a > 1$), negative allometry ($1 > a > 0$), enantiometry ($a < 0$), and isometry ($a = 1$).

In the first PCA approach (3.1) we excluded total length and width characters because of their high influence on the PCA result, since higher values compose the majority

of the summarized variance in PC's (*Mingoti, 2013*), and because of the redundancy between these measurements and the others. We also assessed differences by applying two different substitution algorithms for missing data in PAST, using the default "mean value imputation" option (i.e., missing data are replaced by the column average), and the alternative "iterative imputation" option, which computes a regression upon an initial PCA until it converges to missing data estimations, replacing missing data by such estimations (*Ilin & Raiko, 2010*). The latter is recommended and, after comparing both results, we selected it (see Data S3 to visualize PCA results computed using PAST's default option approach). The second PCA (3.2) was conducted to remove the effect of size (=growth) and perform an exploratory analysis of the shape alone. Six specimens were removed from this analysis because they were broken and the total length or width measures were not measurable.

The univariate analysis was made in order to quantify and describe the variation of the characters set in *Bauruemys elegans* skull, using the assumption that the sample is representative of a single population. The linear regression analyses allowed us to make inferences about osteological shape change related to size change, i.e., related to growth, by assuming that bigger specimens are older than smaller ones. This approach is, therefore, a study of allometry (*sensu Huxley & Teissier, 1936*; *Huxley, 1950*; *Gould, 1966*; *Gould, 1979*; *Somers, 1989*; *Futuyma, 1993*) and the assumption of correlation between size and aging is based on continuous growth to be common on extant turtles (*Klinger & Musick, 1995*; *Shine & Iverson, 1995*; *Congdon et al., 2003*). Since the use of a parametric statistic was infeasible due to the nature of the sample (i.e., a small dataset that do not show homoscedasticity and normality), the PCAs were used to search for a structure of the data that matches to the pattern found by *Romano & Azevedo (2007)* using shell characters (i.e., all individuals plotted inside the 95% confidence ellipse). If the pattern observed is similar to previous morphometric and taphonomic inferences, then the variation is not enough to assume that the sample represents different populations of *Bauruemys elegans* or a different species (see 'Taxonomic considerations on the sample'). In other words, since a parametric test is not feasible with statistical confidence, the lack of structure in the PC plots were herein interpreted as a failure to falsify the single population hypothesis. All principal components were, therefore, analyzed but we present only those with higher variance.

## RESULTS

### Descriptive analysis

The results of the descriptive statistics are summarized in Table 2. As expected, values of total length and width (TLS and WLS) were the most variable among all measurements, because the variation scale in these characters is greater than in others measurements. Characters of the bones forming the upper temporal fossa (i.e., PA, QJ, SQ, QU and OP) had great variation, with the parietal being the most variable in length (SD = 6.45) and the least variable in width (SD = 2.94), whereas quadratojugal obtained the smallest variation in length (SD = 2.38) and the greatest in width (SD = 4.03). Among the characters of the bones forming the lower temporal fossa (i.e., JU, MX, PO, PT and PAL), the variation

**Table 2 Descriptive statistics of all data.** Descriptive statistics of the three sorts of characters analyzed (total length and width, comprised measurements, and proportions of the measurements), including mean values (Mean), median values (Median), standard deviation values (SD), number of entries (N), and maximum and minimum values (Max–Min). All measurements are expressed in millimeters, except un-scaled proportions between two measurements.

| | Characters | Vector[a] | N | Mean | Median | SD | Min–Max |
|---|---|---|---|---|---|---|---|
| Total length and width | TLS | 38–39 | 12 | 63.02 | 63.44 | 10.43 | 50.3–82.15 |
| | TWS | – | 15 | 63.08 | 58.93 | 11.91 | 48.39–94.27 |
| Comprised measurements | LPF | 1–4 | 15 | 8.35 | 8.31 | 1.69 | 4.35–10.94 |
| | LFR | 4–7 | 18 | 12.16 | 12.32 | 2.08 | 9.06–15.59 |
| | LPA | 7–12 | 12 | 28.88 | 27.36 | 6.45 | 20.54–43.80 |
| | LVO | 26–27 | 10 | 6.67 | 6.84 | 1.95 | 3.06–9.79 |
| | LPAL | 27–29 | 13 | 6.91 | 6.22 | 2.33 | 3.42–11.57 |
| | LPT | 29–30 | 19 | 11.72 | 11.94 | 2.42 | 6.95–17.99 |
| | LBS | 30–32 | 20 | 12.76 | 12.57 | 1.77 | 9.71–16.21 |
| | LBO | 32–38 | 13 | 14.16 | 13.38 | 2.12 | 11.13–18.28 |
| | LMX | 11–24 | 18 | 18.49 | 18.31 | 4.11 | 12.39–25.68 |
| | LJU | 10–14 | 14 | 12.42 | 12.32 | 3.28 | 4.46–17.22 |
| | LQJ | 13–18 | 6 | 11.15 | 10.66 | 2.38 | 8.26–14.45 |
| | LQU | 19–25 | 14 | 19.83 | 19.35 | 3.51 | 15.21–26.30 |
| | LPO | 6–13 | 17 | 17.54 | 15.72 | 4.12 | 11.51–24.59 |
| | LSQ | 20–21 | 11 | 11.71 | 11.08 | 3.07 | 8.24–16.57 |
| | WPF | 4–5 | 18 | 7.17 | 7.15 | 1.66 | 3.97–11.27 |
| | WFR | 7–8 | 18 | 10.55 | 10.61 | 1.88 | 7.02–13.55 |
| | WPA | 12–16 | 12 | 22.53 | 22.94 | 2.94 | 17.41–26.85 |
| | SMX | 11–11 | 15 | 47.85 | 46.35 | 7.63 | 39.24–66.10 |
| | WVO | 28–28 | 10 | 4.01 | 3.74 | 1.38 | 2.43–7.23 |
| | WCO | 28–34 | 9 | 7.00 | 6.61 | 1.39 | 5.23–9.10 |
| | WPAL | 29–35 | 14 | 18.08 | 18.23 | 2.37 | 15.24–21.50 |
| | WBS | 33–33 | 19 | 15.35 | 14.71 | 2.19 | 12.07–20.05 |
| | WMX | 10–11 | 16 | 9.80 | 9.84 | 2.24 | 6.48–14.27 |
| | WJU | 14–15 | 7 | 7.26 | 7.28 | 2.19 | 4.11–10.14 |
| | WQJ | 16–25 | 7 | 16.35 | 17.81 | 4.03 | 9.91–21.21 |
| | WPO | 13–14 | 16 | 5.15 | 5.00 | 1.83 | 2.73–9.05 |
| | WOP | 20–22 | 14 | 11.41 | 10.96 | 3.54 | 7.78–17.73 |

| Characters | N | Mean | Median | SD | Min–Max |
|---|---|---|---|---|---|
| LPF/TLS | 9 | 0.13 | 0.13 | 0.04 | 0.05–0.19 |
| LFR/TLS | 11 | 0.19 | 0.18 | 0.02 | 0.17–0.22 |
| LPA/TLS | 8 | 0.51 | 0.49 | 0.08 | 0.45–0.65 |
| LVO/TLS | 8 | 0.11 | 0.12 | 0.03 | 0.06–0.15 |
| LPAL/TLS | 10 | 0.11 | 0.11 | 0.03 | 0.06–0.17 |
| LPT/TLS | 12 | 0.18 | 0.18 | 0.02 | 0.13–0.22 |
| LBS/TLS | 12 | 0.21 | 0.21 | 0.02 | 0.17–0.24 |
| LBO/TLS | 11 | 0.24 | 0.24 | 0.02 | 0.21–0.26 |

**Table 2** (*continued*)

| | Characters | *N* | Mean | Median | SD | Min–Max |
|---|---|---|---|---|---|---|
| Proportions of the measurements | LMX/TLS | 11 | 0.29 | 0.28 | 0.06 | 0.17–0.38 |
| | LJU/TLS | 8 | 0.21 | 0.21 | 0.05 | 0.15–0.29 |
| | LQJ/TLS | 5 | 0.18 | 0.16 | 0.05 | 0.14–0.25 |
| | LQU/TLS | 10 | 0.30 | 0.30 | 0.04 | 0.23–0.37 |
| | LPO/TLS | 11 | 0.29 | 0.29 | 0.03 | 0.23–0.35 |
| | LSQ/TLS | 7 | 0.19 | 0.20 | 0.05 | 0.12–0.24 |
| | WPF/TWS | 13 | 0.12 | 0.12 | 0.02 | 0.08–0.15 |
| | WFR/TWS | 13 | 0.17 | 0.17 | 0.02 | 0.14–0.21 |
| | WPA/TWS | 10 | 0.37 | 0.37 | 0.05 | 0.29–0.44 |
| | SMX/TWS | 12 | 0.75 | 0.76 | 0.06 | 0.67–0.86 |
| | WVO/TWS | 7 | 0.09 | 0.07 | 0.02 | 0.04–0.09 |
| | WCO/TWS | 7 | 0.11 | 0.12 | 0.02 | 0.09–0.13 |
| | WPAL/TWS | 9 | 0.29 | 0.29 | 0.02 | 0.27–0.32 |
| | WBS/TWS | 12 | 0.24 | 0.24 | 0.02 | 0.22–0.28 |
| | WMX/TWS | 12 | 0.16 | 0.15 | 0.04 | 0.08–0.24 |
| | WJU/TWS | 6 | 0.12 | 0.13 | 0.05 | 0.05–0.17 |
| | WQJ/TWS | 7 | 0.29 | 0.30 | 0.08 | 0.16–0.37 |
| | WPO/TWS | 12 | 0.08 | 0.08 | 0.02 | 0.06–0.13 |
| | WOP/TWS | 11 | 0.18 | 0.17 | 0.04 | 0.13–0.23 |

**Notes.**
SD, standard deviation values; N, number of entries; Max–Min, maximum and minimum values.
[a]straight line between two landmarks used to trace linear measurements (see Fig. 2 to visualize the landmarks).

in length was in general greater than in width. Postorbital and maxilla had almost the same variation in length (SD = 4.12 and SD = 4.11, respectively); WPO had the smallest variation within the group of bones forming the lower temporal fossa (SD = 1.83); and the stretch of the maxilla had the greatest variation (SD = 7.63) of all characters measured. Characters of the other bones had smaller values than the aforementioned bones, with the exception of WPO which was smaller than LFR (SD = 2.08), LVO (SD = 1.95), LBO (SD = 2.12),WFR (SD = 1.88) and WBS (SD = 2.19).

## Allometric analysis

Among all comprised measurements, three were enantiometric (LPF, WJU and WQJ); five were positively allometric (LPAL, LPT, LPO, WPF and WPO); and the others were negatively allometric. It is also worth to note that two were not isometric (WPF ($a = 1,0074$; $p = 0.0009$) and WOP ($a = 0.98159$; $p = 0.007$)), although presented angular coefficient very close to 1. All regressions are shown in Figs. 3– 5.

## Principal component analysis (PCA)
### *Raw data*

*Replacing missing data with mean values.* By using the "mean value imputation" approach, a total of 70.32% of the variance was comprised by the first three principal components (PC1 = 42.15%; PC2 = 16.82%; PC3 = 11.35%), so that the others were less significant for the analysis by following the broken stick model, and are not presented. We interpreted that PC1 variation is due to size variation because an approach using all characters has shown

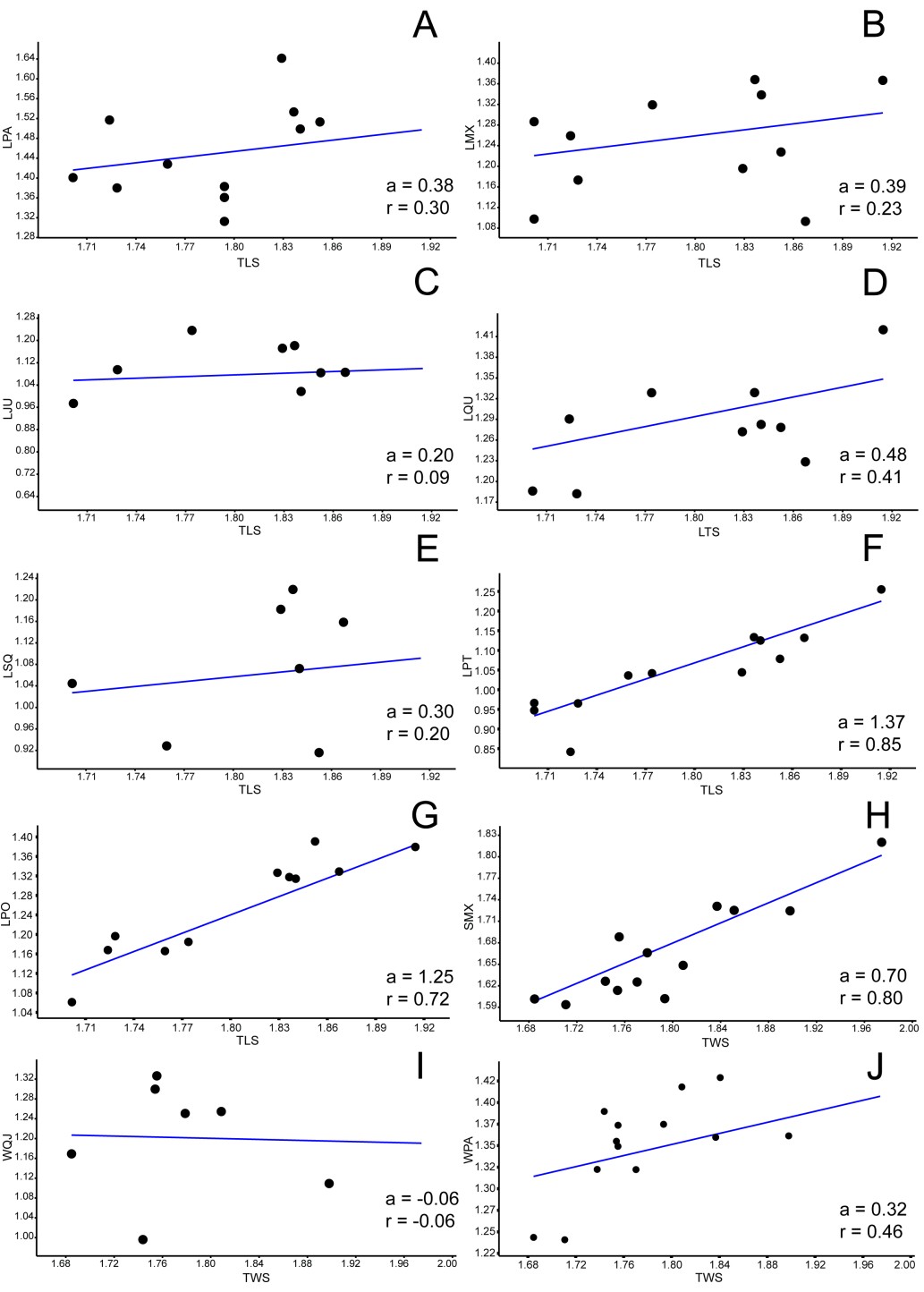

**Figure 3 Allometric graphics: part 1.** Allometries of *Bauruemys elegans* skull bones: (A) length of parietal (LPA), (B) length of maxilla (LMX), (C), length of jugal (LJU), (D) length of quadrate (LQU), (E) length of squamosal (LSQ), (F) length of pterygoid (LPT), (G) length of postorbital (LPO), (H) stretch of maxilla (SMX), (I) width of quadratojugal (WQJ) (J) and width of parietal (WPA). Angular coefficient (a) and coefficient of correlation (r) are shown. Abbreviations: TLS, total length of the skull; TWS, total width of the skull.

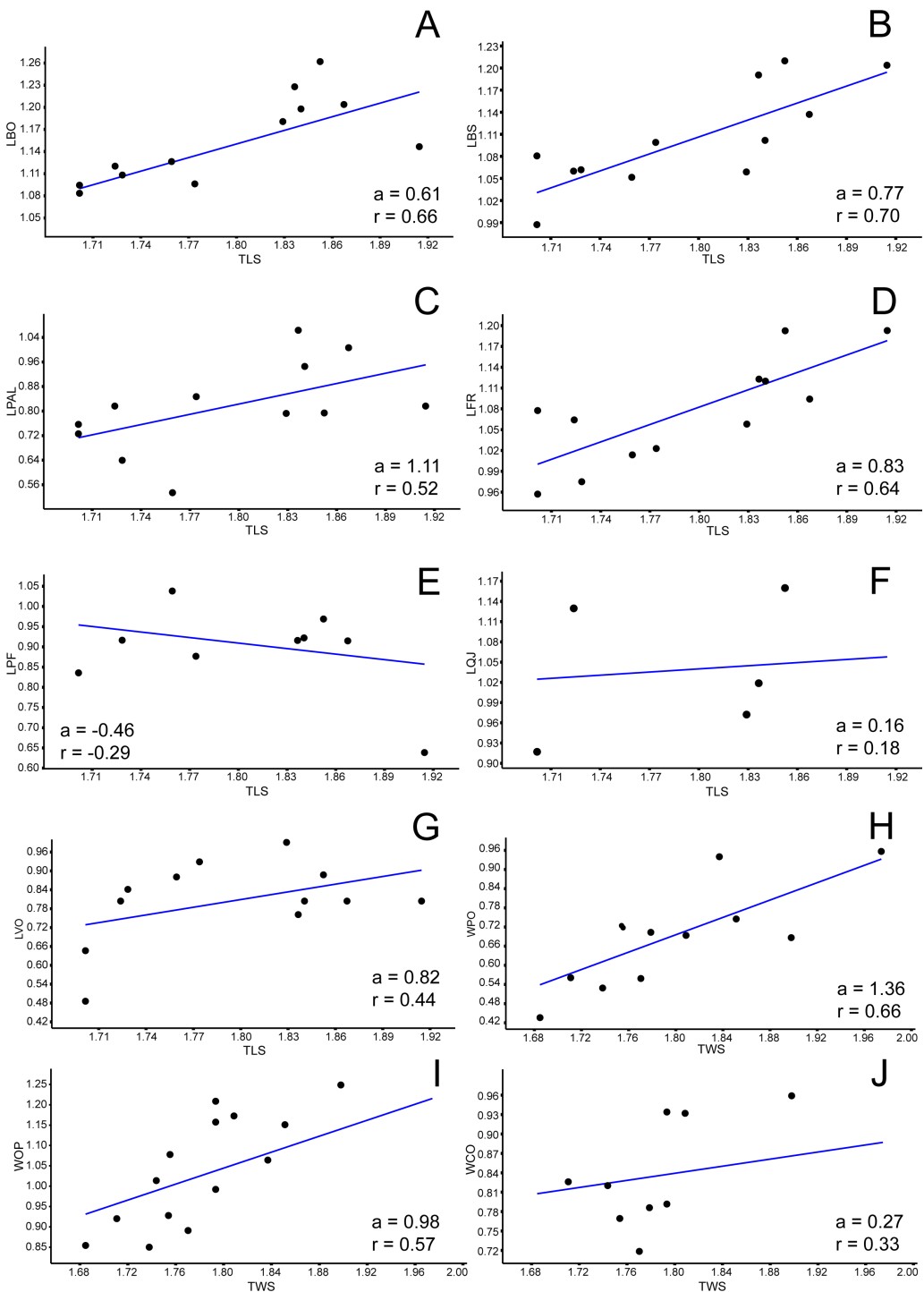

**Figure 4  Allometric graphics: part 2.** Allometries of *Bauruemys elegans* skull bones: (A) length of basioccipital (LBO), (B) length of basisphenoid (LBS), (C), length of palatine (LPAL), (D) length of frontal (LFR), (E) length of prefrontal (LPF), (F) length of quadratojugal (LQJ), (G) length of vomer (LVO), (H) width of postorbital (WPO), (I) width of opisthotic (WOP) (J) and width of choanal (WCO). Angular coefficient (a) and coefficient of correlation (r) are shown. Abbreviations: TLS, total length of the skull; TWS, total width of the skull.

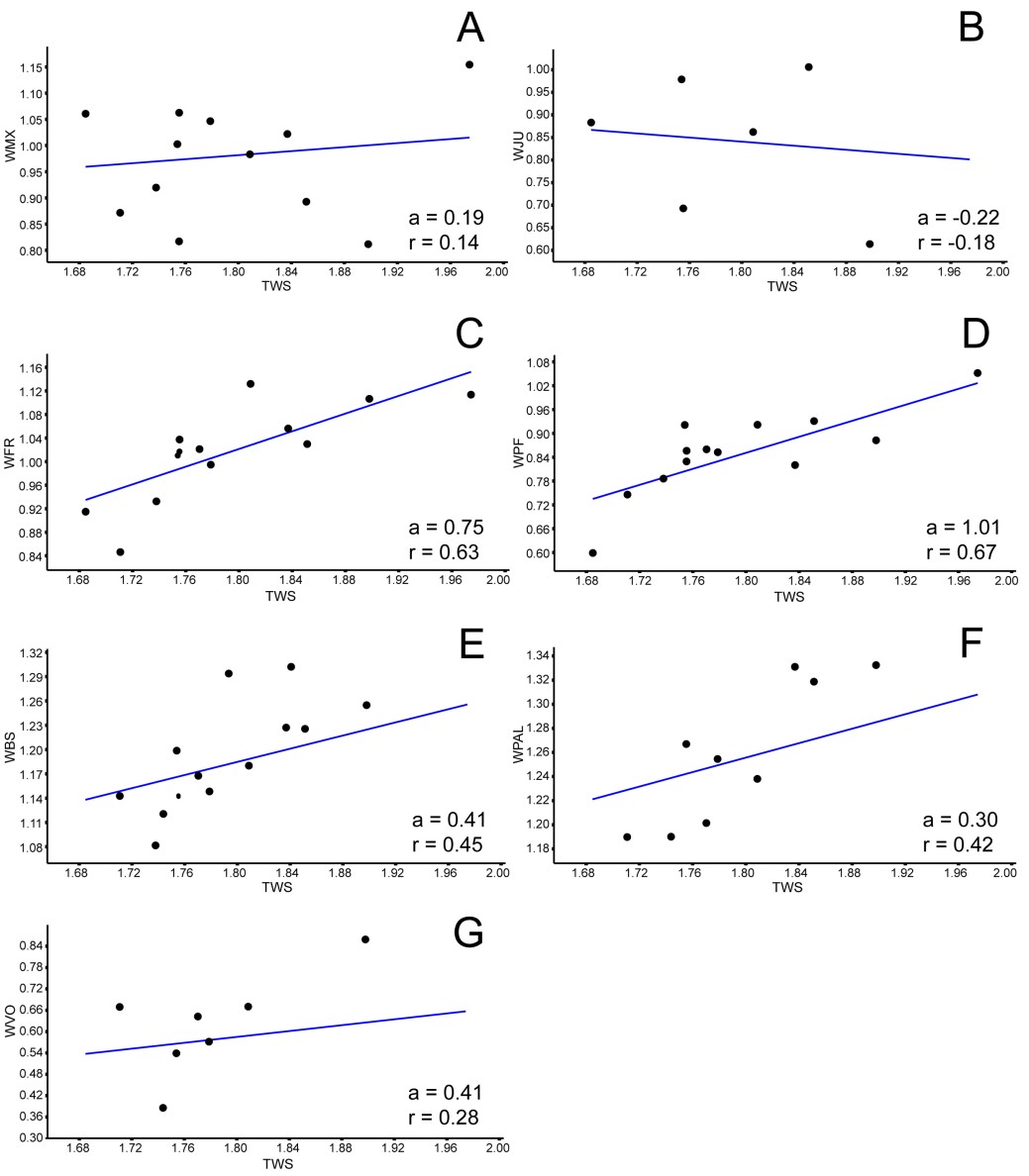

**Figure 5 Allometric graphics: part 3.** Allometries of *Bauruemys elegans* skull bones: (A) width of maxilla (WMX), (B) width of jugal (WJU), (C) width of frontal (WFR), (D) width of prefrontal (WPF), (E) width of basisphenoid (WBS), (F) width of palatine (WPAL) and (G) width of vomer (WVO). Angular coefficient (a) and coefficient of correlation (r) are shown. Abbreviations: TWS, total width of the skull.

a similar plot (see Fig. 6A). PC2 and PC3 seems to represent shape differences between individuals. In all PC individual projections (Figs. 6A and 6B) most of specimens were included inside the 95% confidence ellipse. Two exceptions are MCZ4123 and MN7071-V, which have not been included in the ellipse when PC1 vs. PC2 were considered (Fig. 6A); also the former was outside the ellipse in PC2 vs. PC3 scatter plot (Fig. 6B), indicating form differences of these specimens. However, both specimens have suffered different degrees of crushing due to taphonomic bias and that is likely the reason for this result.

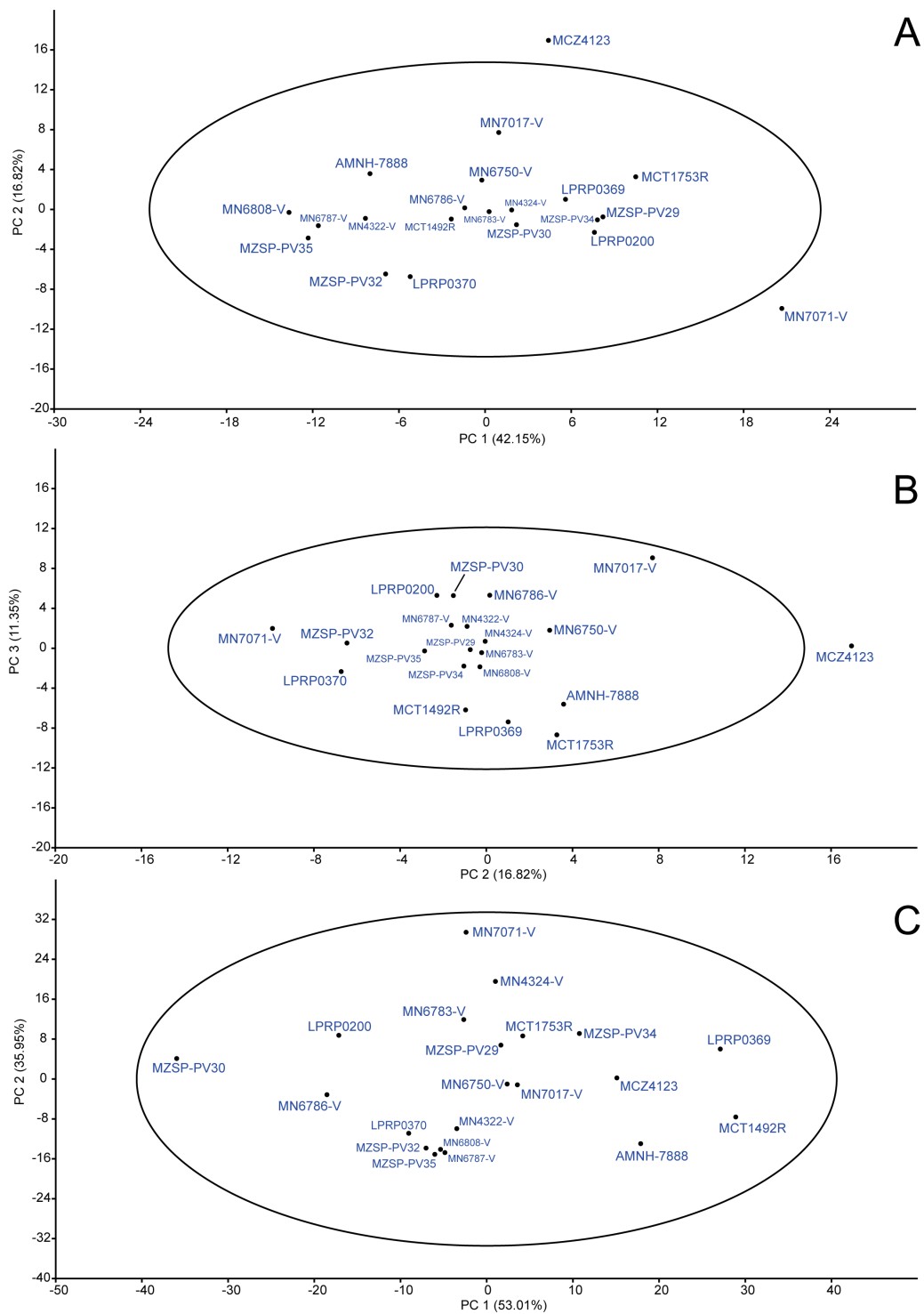

**Figure 6  PCA: raw data.** Principal Components Analysis (PCA) from raw data matrix using mean value substitution approach (A and B) and iterative imputation substitution approach (C) in replacing missing data. The 95% ellipse is given.

**Table 3  PCA loadings: raw data.** Loading values of characters in the raw data matrix related to the first three principal components in PCA, comparing the Mean Value (mv) approach with the Iterative Imputation (ii) approach.

| Char. | PC1 (mv) | PC2 (mv) | PC3 (mv) | PC1 (ii) | PC2 (ii) | PC3 (ii) |
|---|---|---|---|---|---|---|
| LPF | −0.05 | 0.04 | 0.02 | −0.04 | −0.05 | −0.05 |
| WPF | 0.14 | 0.02 | 0.05 | −0.001 | 0.12 | 0.08 |
| LFR | 0.19 | −0.01 | −0.09 | 0.02 | 0.14 | −0.04 |
| WFR | 0.17 | 0.10 | −0.02 | 0.01 | 0.13 | −0.001 |
| LPA | 0.27 | 0.74 | 0.10 | 0.89 | 0.04 | 0.11 |
| WPA | 0.12 | 0.17 | −0.01 | 0.22 | 0.16 | 0.06 |
| SMX | 0.66 | −0.45 | −0.22 | 0.01 | 0.59 | −0.34 |
| LVO | 0.05 | 0.07 | 0.03 | −0.02 | 0.11 | 0.01 |
| WVO | 0.04 | 0.03 | −0.07 | 0.02 | 0.09 | −0.11 |
| WCO | 0.05 | 0.04 | −0.07 | 0.03 | 0.12 | −0.08 |
| LPAL | 0.08 | 0.04 | 0.06 | 0.04 | 0.13 | 0.27 |
| WPAL | 0.15 | 0.02 | −0.09 | 0.03 | 0.23 | −0.05 |
| LPT | 0.17 | −0.14 | 0.08 | −0.02 | 0.13 | 0.10 |
| LBS | 0.14 | −0.02 | 0.01 | 0.01 | 0.10 | 0.05 |
| WBS | 0.12 | 0.05 | −0.07 | 0.02 | 0.19 | −0.05 |
| LBO | 0.11 | 0.11 | −0.07 | 0.03 | 0.20 | 0.03 |
| LMX | 0.18 | −0.17 | 0.68 | −0.18 | 0.16 | 0.38 |
| WMX | 0.09 | −0.07 | 0.25 | −0.08 | 0.11 | 0.19 |
| LJU | 0.08 | 0.13 | 0.30 | −0.14 | 0.19 | 0.25 |
| WJU | −0.01 | 0.02 | 0.10 | 0.01 | −0.01 | 0.21 |
| LQJ | 0.04 | −0.05 | −0.04 | −0.16 | 0.18 | −0.11 |
| WQJ | 0.03 | 0.07 | 0.29 | −0.11 | 0.17 | 0.42 |
| LQU | 0.18 | −0.13 | 0.32 | −0.13 | 0.21 | 0.18 |
| LPO | 0.36 | 0.19 | −0.13 | 0.03 | 0.29 | 0.02 |
| WPO | 0.11 | −0.04 | 0.05 | −0.01 | 0.10 | 0.04 |
| WOP | 0.21 | 0.15 | −0.23 | 0.06 | 0.30 | −0.24 |
| LSQ | 0.07 | 0.19 | 0.11 | 0.16 | 0.02 | 0.43 |

**Notes.**
Char, characters; mv, Mean Value approach; ii, Iterative Imputation approach..

In PC1' loadings (Table 3), only two characters were negatively related (LPF and WJU); SMX, LPA and LPO loadings were the highest related ($L = 0.69$; $L = 0.27$; $L = 0.36$, respectively); and the rest of characters obtained intermediate values (e.g., LPT ($L = 0.17$), LMX ($L = 0.18$), WOP ($L = 0.21$)). PC2 has shown a high relation with character LPA ($L = 0.77$), showing possible changes in shape in this region, and a negative loading for SMX ($L = -0.38$), whereas the others had no significant scores. The last considered principal component (=PC3), showed high correlations with bones in both lateral and posterior emarginations of the skull (LMX ($L = 0.68$), WMX ($L = 0.25$), LJU ($L = 0.30$), WQJ ($L = 0.29$) and LQU ($L = 0.32$)) and, as the results in PC2, allows inferences in shape changes of these regions.

*Replacing missing data with regression estimation.* The alternative missing data approach (i.e., "iterative imputation"; Fig. 6C) generated two principal components which comprised 88.96% of the total variance (PC1 = 53.01%; PC2 = 35.95%). In contrast with the previous approach, PC1 was interpreted as representing shape and PC2 reflected size variations. In addition, all specimens were included inside the 95% ellipse in PC1vs.PC2 scatter plot. The specimen MN7017-V, interestingly, was excluded from the ellipse when considering PC2 vs. PC3, but the percentage of variance represented by PC3 is too low (PC3 = 3.28%) to assume any difference from the others individuals. We agree with *Ilin & Raiko (2010)* and prefer to choose the iterative imputation approach for dealing with missing entries (see discussion on 'The single population hypothesis'). Then, discussions concerning the form variation in our data are related to PCA analysis using iterative imputation.

In PC1 loadings (Table 3), LPA, WPA and LSQ were the highest positively related characters ($L = 0.89$; $L = 0.22$; $L = 0.16$, respectively), whereas LMX, LJU, LQJ, WQJ and LQU were the highest negatively related characters ($L = -0.18$; $L = -0.14$; $L = -0.16$; $L = -0.11$; $L = -0.11$; $L = -0.13$, respectively). Only two characters were negative for PC2 (LPF and WJU), whereas the rest of the coefficients were positive. Among them, SMX was the highest ($L = 0.59$); WPAL, WBS, LBO, LJU, LQU, LPO and WOP obtained intermediate scores ($L = 0.23$; $L = 0.19$; $L = 0.20$; $L = 0.19$; $L = 0.21$; $L = 0.29$; $L = 0.30$, respectively); the others were less related (e.g., LPA ($L = 0.04$), LPT ($L = 0.13$) and WPO ($L = 0.10$)). In general, the values indicate that in *B. elegans* most changes occur in bones of both lateral and temporal emargination.

### Shape characters (proportions)

*Replacing missing data with mean values.* When applying "mean value imputation", 53.99% of the variance were comprised by the first two principal components (PC1 = 35.29%; PC2 = 18.70%), both corresponding to shape, as all units of measurements were removed through the division of characters before carrying out the analysis. All specimens were comprised into the 95% confidence ellipse (Fig. 7A).

The first PC was positively related to the loadings values of LPA/TLS ($L = 0.28$), LMX/TLS ($L = 0.38$), LQU/TLS ($L = 0.27$), WPA/TWS ($L = 0.23$), SMX/TWS ($L = 0.38$), WMX/WTS ($L = 0.35$), WQJ/TWS ($L = 0.48$); the most negative values were LPO/TLS ($L = -0.16$) and WOP/TWS ($L = -0.13$). The second PC was positively related to LPA/TLS ($L = 0.66$), WPA/TWS ($L = 0.32$) WOP/TWS ($L = 0.27$), and negatively to LMX/TLS ($L = -0.50$) (see Table 4 for all loading values). It is interesting to note that most of highly-related proportions were in reference to bones associated either with feeding *apparatus* (squamosal, parietal, quadratojugal and jugal) or catching food and trituration surface (maxilla).

*Replacing missing data with regression estimation.* The "iterative imputation" substitution model of missing data explained 77.35% of the variance comprised by two principal components (PC1 = 45.49%; PC2 = 31.86), both representing shape. All specimens were included in the confidence ellipse (Fig. 7B), thus shape differences do not indicate possible different populations or species.

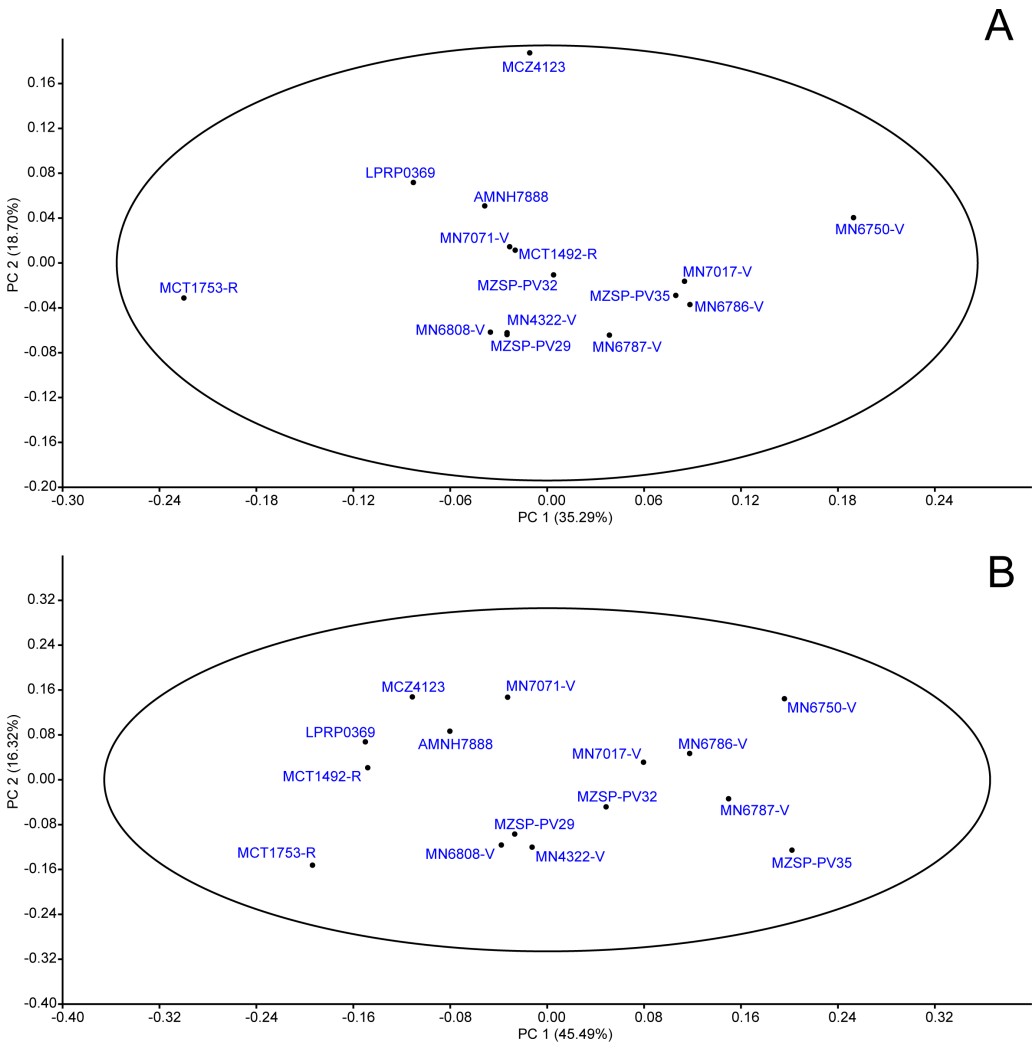

**Figure 7** **PCA: proportions data.** Principal Components Analysis (PCA) from proportions data matrix using mean value substitution approach (A) and iterative imputation substitution approach (B) in replacing missing data. The 95% ellipse is given.

PC1 was highly related to LMX/TLS ($L = 0.48$), LJU/TLS ($L = 0.16$), LQJ/TLS ($L = 0.21$), LQU/TLS ($L = 0.28$), LSQ/TLS ($L = 0.20$), SMX/TWS ($L = 0.33$), WMX/TWS ($L = 0.30$), WJU/TWS ($L = 0.26$) and WQJ/TWS ($L = 0.41$), which represent the highest values, as well as bones constituting both lateral and posterior emargination. Conversely, PC2 was mostly represented by LPA/TLS ($L = 0.67$), LSQ/TLS ($L = 0.34$) and WPA/TWS ($L = 0.33$) (see Table 4). These loadings represent shape changes in regions of the skull that are associated with muscles' attachments as well as trituration surfaces (see below).

## DISCUSSION

### The single population hypothesis

In this section, we discuss the single population hypothesis considering two fronts, one underlied on the taphonomy of the Tartaruguito locality, and another on the possibility of

**Table 4 PCA loadings: proportion data.** Loading values of characters in the proportions data matrix related to the first two principal components in PCA, comparing the Mean Value (mv) approach with the Iterative Imputation (ii) approach.

| Char. | Pc1 (mv) | Pc2 (mv) | Pc1 (ii) | Pc2 (ii) |
|---|---|---|---|---|
| LPF/TLS | 0.003 | −0.13 | 0.11 | −0.30 |
| LFR/TLS | 0.001 | −0.04 | 0.03 | −0.02 |
| LPA/TLS | 0.28 | 0.66 | −0.13 | 0.67 |
| LVO/TLS | −0.002 | 0.05 | −0.03 | −0.02 |
| LPAL/TLS | 0.08 | 0.02 | 0.07 | 0.12 |
| LPT/TLS | −0.05 | −0.10 | −0.02 | −0.01 |
| LBS/TLS | 0.03 | −0.17 | 0.11 | −0.10 |
| LBO/TLS | −0.02 | −0.04 | 0.01 | −0.04 |
| LMX/TLS | 0.38 | −0.43 | 0.48 | −0.18 |
| LJU/TLS | 0.16 | 0.01 | 0.16 | 0.14 |
| LQJ/TLS | 0.06 | −0.09 | 0.21 | −0.17 |
| LQU/TLS | 0.27 | −0.07 | 0.28 | 0.05 |
| LPO/TLS | −0.16 | 0.13 | −0.18 | 0.03 |
| LSQ/TLS | 0.16 | 0.23 | 0.20 | 0.34 |
| WPF/TWS | 0.07 | 0.09 | −0.001 | 0.11 |
| WFR/TWS | 0.07 | 0.13 | 0.02 | 0.05 |
| WPA/TWS | 0.23 | 0.32 | 0.08 | 0.33 |
| SMX/TWS | 0.38 | −0.12 | 0.33 | −0.01 |
| WVO/TWS | −0.05 | −0.04 | −0.04 | −0.10 |
| WCO/TWS | −0.04 | 0.07 | −0.11 | 0.04 |
| WPAL/TWS | 0.04 | −0.07 | 0.04 | −0.003 |
| WBS/TWS | 0.03 | −0.05 | 0.02 | −0.03 |
| WMX/TWS | 0.35 | −0.05 | 0.30 | 0.03 |
| WJU/TWS | 0.18 | 0.01 | 0.26 | 0.19 |
| WQJ/TWS | 0.48 | −0.003 | 0.41 | 0.20 |
| WPO/TWS | 0.02 | 0.01 | −0.01 | 0.07 |
| WOP/TWS | −0.13 | 0.27 | −0.21 | 0.09 |

**Notes.**

Char, characters; mv, Mean Value approach; ii, Iterative Imputation approach.

the skull variation represent one or more specimens of species *Roxochelys wanderleyi* in the sample, a shell-only species also found at the site.

### The depositional context at the "Tartaruguito" site

The depositional environment at the Pirapozinho site is well-known from previous studies, which point out to seasonal floods in which turtles might have gathered in water bodies for foraging, followed by droughts that caused their death (*Soares et al., 1980*; *Fulfaro & Perinotto, 1996*; *Fernandes & Coimbra, 2000*; *Henriques et al., 2002*; *Henriques et al., 2005*; *Suárez, 2002*; *Bertini et al., 2006*; *Henriques, 2006*). This is, consequently, a case of several seasonal non-selective death events, with individuals representing semaphoronts connected temporally (between generations), thus comprising a single population (agreeing with *Futuyma*'s, *1993* population definition and used by *Romano & Azevedo, 2007*). We failed to

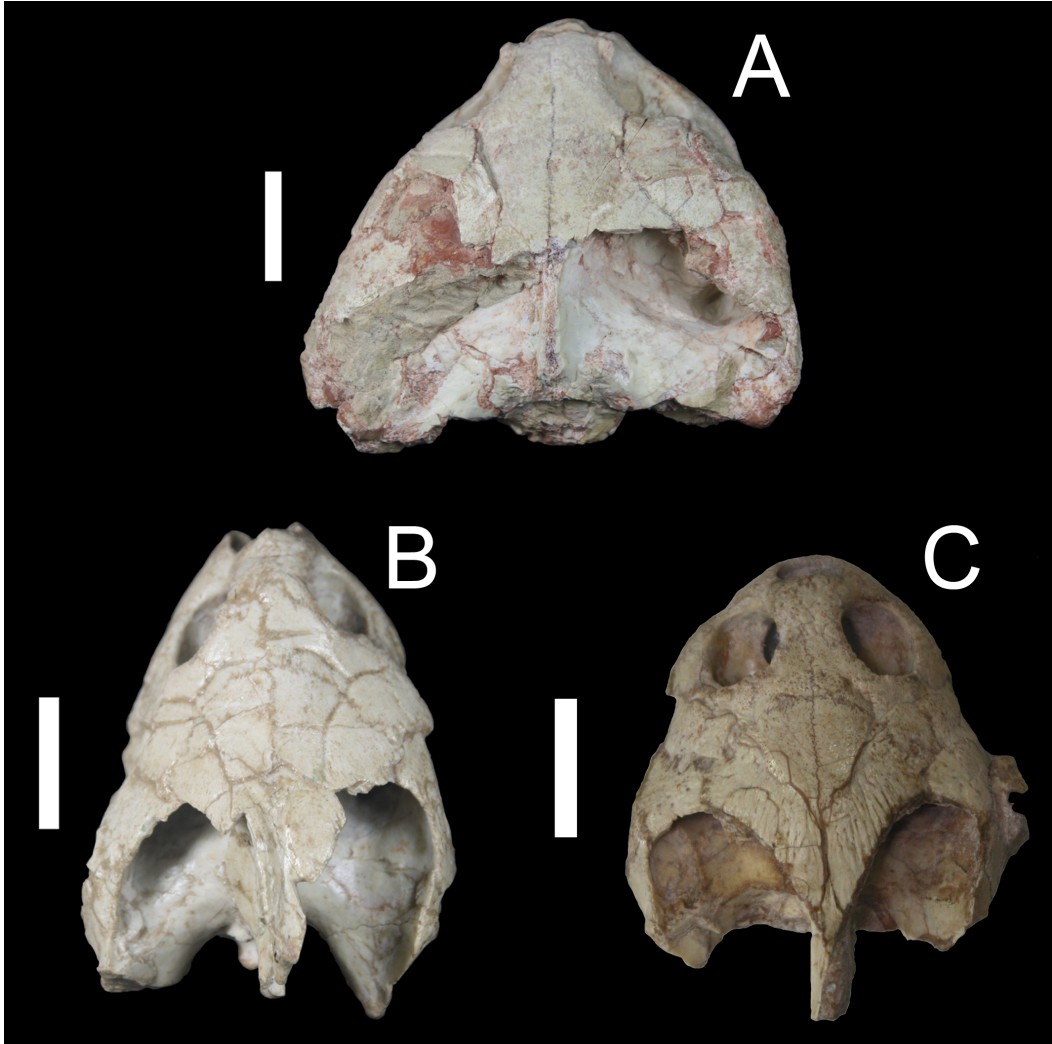

**Figure 8** **Comparison of a taphonomically altered skull with two well-preserved skulls of *Bauruemys elegans*, showing the cheek morphologies observed.** *Bauruemys elegans* specimens in dorsal view showing the largest MN7071-V specimen (A) in contrast with two smaller, well-preserved narrow-cheeked MN7017-V (B) and wide-cheeked MN4322-V (C) specimens. MN7071-V (A) is larger due to vertical crushing in the mediocaudal portion of the skull, resulting in artificial wide-cheeked morphology. In other specimens, such a taphonomic effect is not observed, indicating that both narrow- (B) and wide-cheeked (C) morphologies are naturally present in *B. elegans*.

disprove the null hypothesis that all individuals belong to a same population of *Bauruemys elegans*, agreeing with *Romano & Azevedo (2007)* conclusion using post-cranium data.

### Taxonomic considerations on the sample

Many skulls sampled show taphonomic effects, such as cracks and crushing (Fig. 8). For instance, MN7071-V is notably the largest specimen of the sample and is represented in the uppermost positive side of the size-related PC2 axis (Fig. 6C). Although it is indeed a big specimen, it was clearly a taphonomic effect (crushing) that caused it to be larger than it really was (Fig. 8A). On the other hand, *Bertini et al. (2006)* indicated that turtle

bodies have suffered little transportation or crushing in Tartaruguito site. We agree with this taphonomical interpretation of the site, as most specimens do not show huge breaks (Figs. 8B and 8D) that could cause misinterpretation of the morphometric results (the case of MN7071-V is an exception in our sample in this respect).

Another aspect is the presence of polymorphism in *B. elegans*. *Romano (2008)* presented an unusual carapace for the specimen MN7017-V, as having a seventh neural bone, differing from the diagnostic number of six neurals for this species, and with the diagnostic four-squared second neural bone not contacting first costals (*Suárez, 1969b*; *Suárez, 1969c*; *Kischlat, 1994*; *Gaffney et al., 2011*). The morphometric analysis performed by *Romano (2008)*, using only shells, did not reveal significant statistical differences between MN 7017-V and other *B. elegans* specimens. We have included the MN7017-V skull in our analysis, and there was no variation to state anything apart from *Romano*'s (*2008*) conclusion that it is probably a polymorphic *B. elegans* specimen (Fig. 6C). Still, we reevaluated this skull and found the diagnostic characters for *B. elegans*. Therefore, all skulls included in our study belong to the same species (i.e., *B. elegans*).

Among the five valid fossil turtle species found throughout the Bauru Basin, only two have been collected at the Pirapozinho site so far (*Romano et al., 2013*). The first is *B. elegans*, which is recognized by both skull and shell materials; the second is *Roxochelys wanderleyi* Price, 1953 based only on shell material (*Price, 1953*; *De Broin, 1991*; *Oliveira & Romano, 2007*; *Romano & Azevedo, 2007*; *Gaffney et al., 2011*; *Romano et al., 2013*). So far, none *R. wanderleyi* with skull-shell associated body parts were collected, and thus we cannot claim that the skulls found at Tartaruguito site belong to this species until a skull-shell *R. wanderleyi* specimen be found, since all skulls analyzed here can be safely identified as belonging to *B. elegans*.

## Ontogenetic changes in *B. elegans* skull

Once we have assessed that all specimens belong to the same species and are likely from the same population, we are able to discuss the skull variation in the sample assuming as due to inter-populational variety. For the sake of organization, we divided the discussion into two parts, based on the anatomical regions of the turtle skull: upper temporal fossa and lower temporal fossa, following *Schumacher (1973)*, *Gaffney (1979)* and *Gaffney, Tong & Meylan (2006)*. We have chosen this organization because the bones we found most associated with the principal components in the two PCA analyses constitute these two regions and are generally involved in aspects of the feeding mechanisms of turtles, either as muscles attachments or forming triturating surfaces.

### Bones of the upper temporal fossa and skull roofing

The temporal emargination of podocnemidid turtles is formed by the dorsal, horizontal plate of the parietal, the quadratojugal and the squamosal, with no contribution of the postorbital (*Gaffney, 1979*; *Gaffney et al., 2011*). This region (and bones) is associated with the origin of the adductor muscle fibers (m. adductor complex; Figs. 9A and 9B) (*Schumacher, 1973*; *Werneburg, 2011*; *Werneburg, 2012*; *Jones et al., 2012*; *Werneburg, 2013*), which run through *cartilago transiliens* of the *processus trochlearis pterygoidei* of

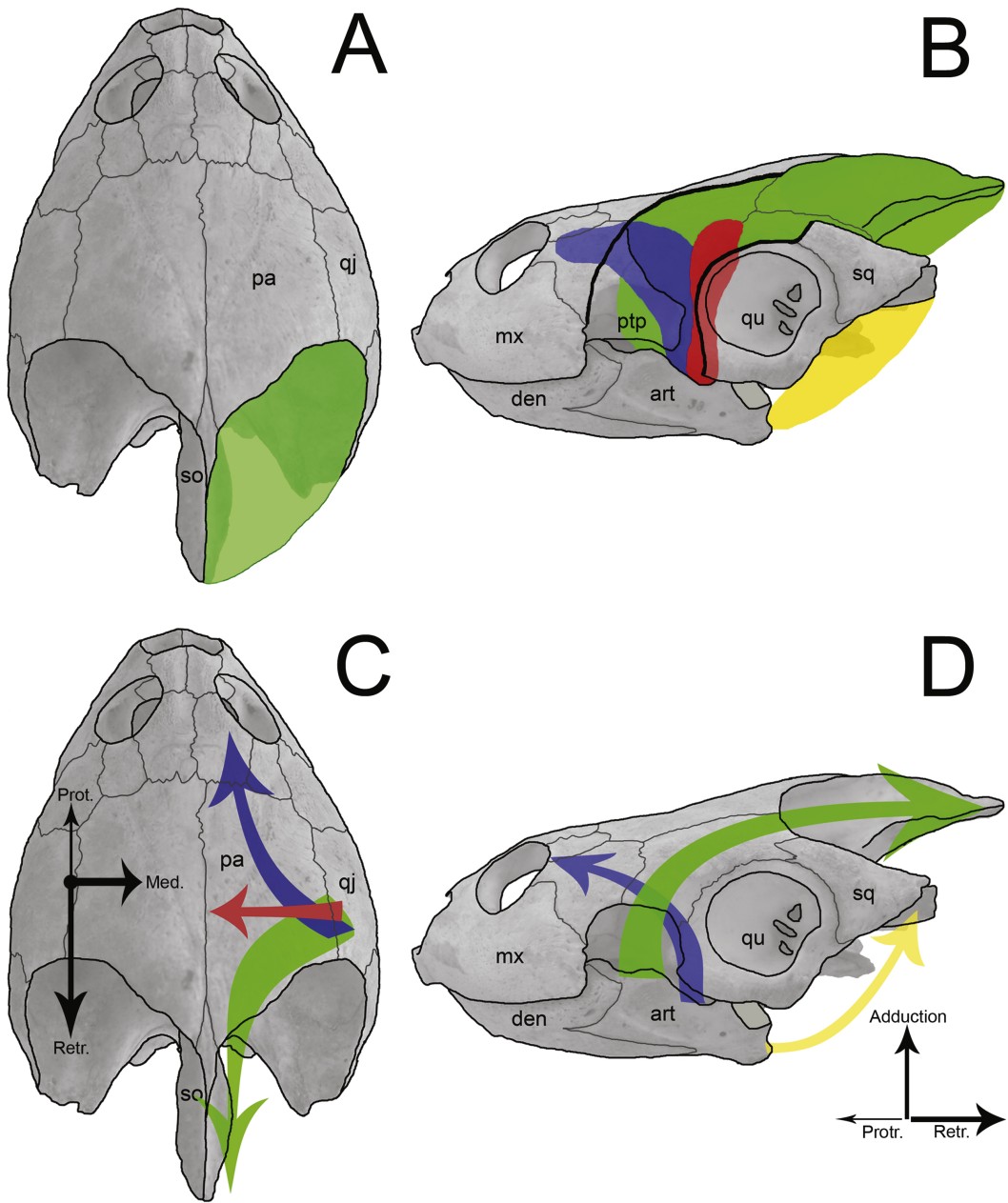

**Figure 9** **Sketch of jaw-closing muscles and its vector forces in *Podocnemis expansa*.** Dorsal (A and C) and left lateral (B and D) view of the skull of *Podocnemis expansa* (MZSP-0038) showing the muscle attachment places (A and B) and the direction vector forces (C and D) during jaw closing. The muscles and vectors of *m. adductor mandibulae externus* (green), *m. adductor mandibulae posterior* (red), *m. pterygoideus* (blue), and *m. depressor mandibulae* (yellow) are sketched. Length and thickness of the arrows indicate the relative forces. Abbreviations: art, articular; den, dentary; mx, maxilla; pa, parietal; ptp, processus trochlearis pterygoidei; qj, quadratojugal; qu, quadrate; so, supraoccipital.

the pterygoid and insert at the coronoid process of the lower jaw (*Schumacher, 1973*; *Gaffney, 1975*; *Gaffney, 1979*; *Lemell, Beisser & Weisgram, 2000*; *Werneburg, 2011*). These muscles promote the closure of the mouth, thus it is reasonable to associate the attachment surface to bite force and the latter to the prey hardness. Yet on the ventral flange of the squamosal originates the *m. depressor mandibulae* (*Schumacher, 1973*; *Gaffney, Tong & Meylan, 2006*; *Werneburg, 2011*; Fig. 9B), which causes the abduction (=opening) of the mandible.

The variation in this area of the skull in turtles was a matter of some studies (e.g., *Dalrymple, 1977*; *Claude et al., 2004*; *Pfaller, Gignac & Erickson, 2011*), which indicated allometric ontogenetic growth patterns of the bones in these regions. These authors were able to identify a high correlation with the increasing of muscle mass and shift in feeding features (*Dalrymple, 1977*; *Pfaller et al., 2010*; *Pfaller, Gignac & Erickson, 2011*). Moreover, there are changes in skull shape associated to the aquatic environment and foraging strategies, as suggested for emydid and testudinoid turtles by *Claude et al. (2004)*. Although these studies focused on hide-necked turtles, the same morphoecological patterns can be applied to side-necked turtles, since there are habitat occupation similarities between side-necked and hide-necked turtles with implications to the skull morphology due to morphofunctional constraints (*Schumacher, 1973*; *Lemell, Beisser & Weisgram, 2000*), besides the adaptive selection regarding fresh water feeding strategies (see *Lauder & Prendergast, 1992*, *Aerts, Van Damme & Herrel, 2001* and *Van Damme & Aerts, 1997* for feeding strategies in freshwater turtles).

The high variance and positive allometric growth of the parietal (LPA: $a = 0.38$; WPA: $a = 0.32$), quadratojugal (LQJ: $a = 0.16$; WQJ: $a = -0.06$) and squamosal (LSQ: $a = 0.30$) lead to an increase in temporal emargination and, consequently, a greater area for attachment of the *m. adductor mandibulae externus*. The consequence of this would be the generation of large forces and high velocities during the fast closing phase of an aquatic feeder, as seen in *Pelusios castaneus* (*Lemell, Beisser & Weisgram, 2000*), and even a more powerful bite for crushing harder prey, as seen in *Sternotherus minor* (*Pfaller, Gignac & Erickson, 2011*). In addition, the lenghten of the squamosal would allow a greater insertion area of the *m. depressor mandibulae* and muscles of the hyobranchial apparatus (e.g., *m. constrictor colli*) (*Schumacher, 1973*; *Gaffney, 1979*; *Claude et al., 2004*; *Gaffney et al., 2011*; *Werneburg, 2011*). The *m. depressor mandibulae* is useful for an increased gape opening speed and the hyobranchial apparatus musculature is involved in backwards water flow generation by the lowering of the hyoid apparatus, two characteristics well reported for other pleurodire turtles as *Chelodina longicollis*, *Chelus fimbriatus* and *Pelusios castaneus* (e.g., *Van Damme & Aerts, 1997*; *Aerts, Van Damme & Herrel, 2001*; *Lemell, Beisser & Weisgram, 2000*; *Lemell et al., 2002*). Moreover, *Claude et al. (2004)* demonstrated that aquatic turtles with suction feeding mode possess longer skulls than terrestrial turtles, the squamosal being most prominent bone involved in this elongation and functionally related to the style of prey capture (=suction) as a support for mandible and hyoid muscles.

Also, *Gaffney et al. (2011)*, in a comparison with other podocnemidid turtles, indicated *B. elegans* as having a "skull relatively wide and flat" (p. 12), which could be observed by the increasing of some bones, specially the postorbital (Figs. 3G and 4H), parietal (Figs.

3A and 3J), quadratojugal (Figs. 3I and 4F) and jugal (Figs. 3C and 5B). Comparing the postorbital allometry (better discussed below) with those of the bones in contact with it in the skull roof (frontal, parietal, jugal and quadratojugal; *Gaffney et al., 2011*), we observe an influence of the positive growth of the former into the others, leading to flattening and widening of the skull.

In a study assessing the bite performance in turtles, *Herrel, O'Reilly & Richmond (2002)* suggested that a higher skull is efficient in promoting stronger bite forces, specially in species which feed on hard prey, but they also pointed out that additions in bite forces may be achieved by "getting longer and larger" skull with no increasing in skull height. Thus, in addition to provide gains in muscle attachment area, by the growing of parietal, quadratojugal and squamosal, leading to a longer skull, a stronger bite and possibly a change in diet along the ontogeny. Also, the allometric growths of most of skull bones, particularly the positive allometry of the postorbital, suggests a more roofed skull in larger adults of *B. elegans* bigger adults. Given the allometric patterns aforementioned, *B. elegans* may have had a wide and flat but long skull, which would have compensated the loss of muscle volume and attachment area caused by widening and flattening the skull (*Herrel, O'Reilly & Richmond, 2002*). Correlations between a more emarginated skull and increases in the volume of the adductor muscles were also explored in a cranial evolutionary framework of stem-turtles by *Sterli & De la Fuente (2010)*.

At last, *Gaffney, Tong & Meylan (2006)* and *Gaffney et al. (2011)* scored a character based upon the contact between quadratojugal and parietal bones (char. 13 of *Gaffney, Tong & Meylan, 2006*; char. 5 of *Gaffney et al., 2011*). They also state that this contact is present in *Hamadachelys* + Podocnemididae clade, with a large quadratojugal (state 1), in contrast to most of other Pelomedusoides (state 0: contact absent, as seen in Pelomedusidae, Araripemydidae and many bothremydids (e.g., Kurmademydini, Cearachelyini and Bothremydini); state 2: contact present with small quadratojugal in some Taphrosphyini, Bothremydidae). Indeed *B. elegans* possess a large quadratojugal, which means that the reduction of the postorbital evolved after *Bauruemys* node of divergence, as confirmed in performed cladistic analyses. However, we found a greater increasing (positive allometry) of the two measurements of the postorbital and this might have influenced the growth of parietal and quadratojugal, as well as the jugal (see below), so that the state 1 seen in *B. elegans* is possibly a consequence of allometric changes. This is easily seen when comparing the enatiometry of the width of the quadratojugal (WQJ: $a = -0.06$) and the slight increasing in the length of this bone (LQJ: $a = 0.16$) with the postorbital measurements. It also could have influenced the growth of the parietal, but to a lesser extent, as seen in the allometries of this bone (LPA: $a = 0.38$; WPA: $a = 0.32$).

When comparing the stem-Podocnemidinura species (i.e., *Brasilemys*, *Hamadachelys*) and stem-Podocnemididae (e.g., *Bauruemys*, *Peiropemys*, *Pricemys* and *Lapparentemys*), with the crown-Podocnemididae (i.e., *Podocnemis* lineage + *Erymnochelys* lineage) (*Gaffney et al., 2011*; Fig. 10), it is clear that an increasing in the parietal-quadratojugal contact has occurred along the podocnemidid lineage, and consequently led to a more roofed and less emarginated skull. We suggest that in *B. elegans* the small contact is due to the positive growth of the postorbital resulting in a more emarginated skull than other podocnemidids,
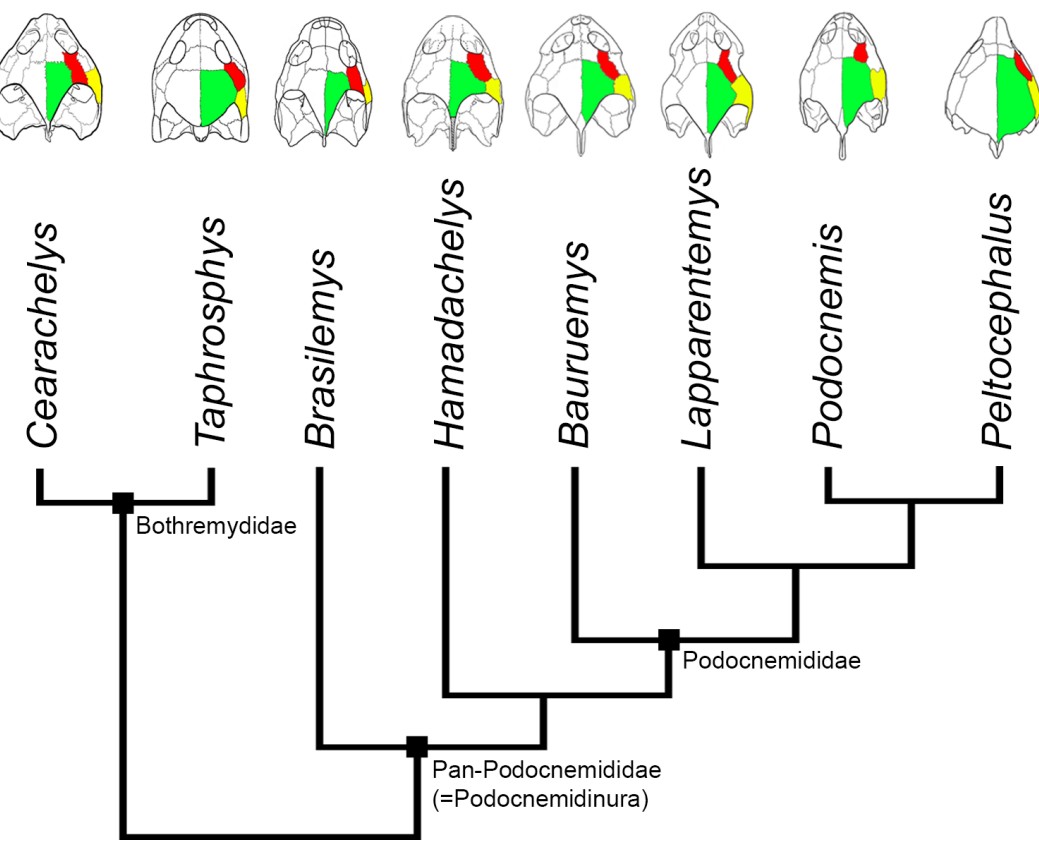

**Figure 10  Evolution of PA-QJ contact and skull roofing in Podocnemidoidea.** Simplified phylogeny of Podocnemidoidea (Bothremydidae + Pan-Podocnemididae) showing the evolution of the contact between parietal (green; PA) and quadratojugal (yellow; QJ), and its relation with the postorbital (red; PO) and skull roofing. Within Bothremydidae, both very emarginated (*Cearachelys placidoi*) and less emarginated (*Taphrosphys congolensis*) skulls are present, showing either no contact (*C. placidoi*) or contact present with small QJ (*T. congolensis*). Within Pan-Podocnemididae, the contact PA-QJ is present and the skull roofing increased from a less roofed condition, found in *Brasilemys josai* and *Hamadachelys*, to a continuous increasingly growing well roofed condition within Podocnemididae, exemplified by *Bauruemys elegans*, *Lapparentemys vilavillensis* and *Podocnemis unifilis*, up to a fully roofed morphology in *Peltocephalus*. *Cearachelys placidoi* and *T. congolensis* modified from *Gaffney, Tong & Meylan (2006)*; *Brasilemys josai* redrawn from *Lapparent de Broin (2000)*; all others skulls modified from *Gaffney et al. (2011)*.

as described by *Gaffney et al. (2011)*. Yet within crown-Podocnemididae this bone suffered the opposite effect (i.e., small growth), showing variations in size and even being absent in some species (e.g., *Podocnemis sextuberculata*, *Ruckes, 1937*; *Gaffney, 1979*; *Gaffney et al., 2011*), though the emargination is still great. On the other hand, in the *Erymnochelys* lineage the postorbitals are large but the quadratojugal and parietal are large as well, leading to a greater contact between these bones and a well-roofed but less emarginated skull, being a reversion in *Bairdemys venezuelensis* and *B. sanchezi* within the *Erymnochelys* lineage (*Gaffney et al., 2011*). Therefore, the increase or decrease in the temporal emargination within Podocnemididae could be due to variation of allometric patterns in bones that form the skull roof, particularly the postorbital, quadratojugal, and parietal, among different

lineages. Given that observation, we speculate that characters related to the form of the aforementioned bones (postorbital, squamosal, and parietal) are potencially more susceptible to homoplasy.

### Bones of the lower temporal fossa

The lower adductor chamber in Pelomedusoides is formed externally and laterally by the jugal and quadratojugal, with the addition of the maxilla in some cases (e.g., *Podocnemis* spp. and *Bairdemys sanchezi*). The well developed cheek emargination, found in most but not all podocnemidid turtles (the exceptions are all within the *Erymnochelys* lineage but *Bairdemys* spp., *Cordichelys antiqua* and *Latentemys plowdeni*), is also part of the adductor chamber (*Gaffney, 1979*; *Gaffney, Tong & Meylan, 2006*; *Gaffney et al., 2011*). Internally and medially, the postorbital, the jugal, and the pterygoid compose the *septum orbitotemporale*, partially separating the *fossa orbitalis* from the *fossa temporalis*; along with the palatine, they aid to support the *processus trochlearis pterygoidei* of the pterygoid (*Gaffney, 1975*; *Gaffney, 1979*; *Gaffney, Tong & Meylan, 2006*). There is a passage medially to the process of the pterygoid and the *septum orbitotemporale*, running from the *fossa orbitalis* to the *fossa temporalis*, the *sulcus palatinopterygoideus*. The palatine and pterygoid form the floor of its passage, whereas the parietal, postorbital and frontal limit its upper portion. In this region, the *m. adductor mandibulae* fibers run through the *processus trochlaris pterygoidei*, and the *m. adductor mandibulae internus* (i.e., *m. pterygoideus* and *pars pseudotemporalis*; Fig. 9B) mostly originates throughout the pterygoid and parietal bones (*Schumacher, 1973*; *Lemell, Beisser & Weisgram, 2000*; *Lemell et al., 2002*; *Werneburg, 2011*). The *m. adductor mandibulae internus* fibers are involved in the jaw-closure system by generating counter forces (protraction) to the *m. adductor mandibulae externus* (retraction) (*Schumacher, 1973*; *Lemell, Beisser & Weisgram, 2000*; *Lemell et al., 2002*; Figs. 9C and 9D).

Variation of the upper temporal fossa has been studied in different turtles, such as various trionychids (*Dalrymple, 1977*) and *Chelydra serpentina* (*Herrel, O'Reilly & Richmond, 2002*). However, few studies report the variation of the lower adductor chamber, although both the upper and lower temporal fossa are anatomically and functionally coupled (*Schumacher, 1973*). *Dalrymple (1977)* identified a positive allometry in the width of the "temporal passageway" in trionychids. This area is related to the cryptodire pulley system (i.e., a *processus trochlearis* formed by the quadrate and opisthotic) and is analogous to the pleurodire pterygoid process, and thus can be comparable functionally (*Gaffney, 1979*). *Herrel, O'Reilly & Richmond (2002)* concluded that the increase of the bite force in turtles is due to either the increased height of the skull, leading to a more open angle of the *processus trochlearis* in relation to skull longitudinal axis, or to enlargement (in width and length) of the skull, because it allows more area for muscle attachment and volume. We observed the same pattern of growth change in *B. elegans*, as evidenced by the positive allometry of the parietal, postorbital, palatine and pterygoid bones. Other features were observed by *Dalrymple (1977)* in trionychids (e.g., height and width of the supraoccipital crest, lengthening of the squamosal crest and a development of a horizontal crest in the parietal) and were correlated to changes in skull shape with a shift in feeding habits, from

softer to harder preys as individuals age. Again, this seems to be the case in *B. elegans*, as evidenced by the positive allometry of the squamosal and parietal bones.

The bones that mainly compose the skull rostrolaterally and the lateral emargination revealed a correlated allometric shape shift. Even so, jugal and maxilla showed small allometric variation (Figs. 4B, 4C, 6A, and 6B). The reduction of the jugal (WJU: $a = -0.23$) and quadratojugal (WQJ: $a = -0.06$) along with the small growth of the maxilla (WMX: $a = 0.19$) demonstrate a decrease in height at the anterior portion of the skull. Because of the contact between jugal and quadratojugal with the postorbital (and its increase; see previous topic), we suggest that the latter would possibly has affected the growth of the former bones. Moreover, the strong development of the postorbital would ultimately affect the width of the maxilla, which in turn would also affect the jugal. In contrast, the lengthen of this bone would be less affected (LMX: $a = 0.39$). In addition, there is a considerable increment in the stretch of maxilla (SMX: $a = 0.70$) (Fig. 3H) leading to a broader rostrum. Yet this could allow a greater area for crushing, as observed by *Kischlat (1994)* for *B. elegans*, but also related to ontogenetic growth (T Mariani, pers. obs., 2016). All these allometric changes indicate that *B. elegans* owns a more flattened and wider skull (*Gaffney et al., 2011*), which could have allowed greater bite forces generation (*Herrel, O'Reilly & Richmond, 2002*).

There are other morphological implications in which the lower adductor chamber bones are involved and that are worth discussing. As previously pointed out, three bones compose the *septum orbitotemporale*: pterygoid, jugal, and postorbital (*Gaffney, 1979*; *Gaffney, Tong & Meylan, 2006*). Together with the palatine, these three bones provide support for the *processus trochlearis pterygoidei*, where upon runs the tendon that connect the m. *adductor externus* complex into the lower jaw (*Schumacher, 1973*; *Gaffney, 1975*; *Gaffney, 1979*; *Lemell, Beisser & Weisgram, 2000*; *Gaffney, Tong & Meylan, 2006*; *Werneburg, 2011*). Nearby the process, many muscle fibers originate or cross towards their insertions points (*Schumacher, 1973*; *Werneburg, 2011*). The temporal emargination at the upper adductor chamber becomes more emarginted during growth. As a consequence, the attachment area for m. *adductor mandibulae externus* increase during aging, potentially generating stronger bite forces. The consequence of this temporal emargination indentation is that the trochlear process would must become more robust to support higher forces. We interpret that the positive allometries of pterygoid (LPT $a = 1.37$), postorbital (LPO $a = 1.25$ and WPO $a = 1.36$), and palatine (LPAL $a = 1.11$) could be a response to this robustness of the trochlerar process during growth. In other words, they would act together by giving more resistance to the area in which the high forces created by the m. *adductor mandibulae externus* are applied. *Gaffney (1979)* suggested this robustness occurs because muscle volume increase and, consequently, higher bite forces, so these three bones would reinforce the *septum orbitotemporale* in order to support and do not break when muscles are contracted. In addition to such reinforcement, the growth of palatine could be associated with a larger area for crushing preys such as mollusks and crustaceans, as pointed out by *Kischlat (1994)*.

The m. *adductor mandibulae internus* and m. *adductor mandibulae posterior* (Fig. 9B), which originate at the quadrate, prootic, pterygoid, palatine, postorbital and the descending

process of the parietal (*Schumacher, 1973*; *Werneburg, 2011*), are important during the jaw-closure phase. The importance of these muscles has been debated for early tetrapods with flat skull and aquatic lifestyle (e.g., Temnospondyli and Lepospondyli; *Frazzetta, 1968*), in which the internal muscle might have assumed the main function of closing the jaw (*Werneburg, 2012*). This also occurs in turtles with flat skulls and with poorly developed *crista supraoccipitalis* (e.g., Chelidae; *Werneburg, 2011*; *Werneburg, 2012*). However, *B. elegans* does not have a skull as flat as chelids, but has a long supraoccipital bone as well as a greater temporal emargination (*Gaffney et al., 2011*), indicating more area and volume available to *m. adductor mandibulae externus* (*Dalrymple, 1977*; *Sterli & De la Fuente, 2010*). The mechanical effects of adductor muscles upon the lower jaw during food capture has been demonstrated in some turtles (*Schumacher, 1973*; *Lemell, Beisser & Weisgram, 2000*; *Lemell et al., 2002*; *Pfaller, Gignac & Erickson, 2011*). These studies agree that besides acting to close the mouth, the *m. adductor mandibulae internus* executes counter protraction forces to the *m. adductor mandibulae externus* retraction forces, while *m. adductors mandibulae posterior* produce medial forces (Figs. 9C and 9D). The contraction of all these muscles together avoid displacements of the mandible and reduce stresses at the articulation (*Schumacher, 1973*; *Lemell, Beisser & Weisgram, 2000*; *Lemell et al., 2002*). The positive allometries of the bones of the lower adductor chamber of *B. elegans*, therefore, may reflect greater resistance for a more robust musculature of *m. adductor mandibulae internus* and *m. adductor mandibulae posterior* in response to higher forces created by external adductors. Besides, these muscles also play the main role in feeding, as proposed for aquatic feeders (*Frazzetta, 1968*; *Werneburg, 2012*), in addition to a larger area between the two tips of the maxilla (i.e., SMX $a = 0.70$) and a flattened skull.

## Feeding changes over ontogeny in *B. elegans*

Changes in skull shape may be due to habitat differences in which terrestrial turtles (e.g., testudinids) possess higher and shorter skulls while aquatic turtles (e.g., emydids) have flatter and longer skulls (*Claude et al., 2004*). The changes in skull shape of turtles along ontogeny have been assessed in living species (*Dalrymple, 1977*; *Pfaller, Gignac & Erickson, 2011*). Generally, it is supported that a diet shift occurs from small soft prey to bigger harder ones, in association with higher, larger and more robust skulls. These, in turn, are more suitable for crushing clams and/or to capture fishes by having a greater gape. The overall aquatic morphology comprises adaptations to suction feeding, which was also discussed by *Herrel, O'Reilly & Richmond (2002)*, and could be the case of *B. elegans*. Firstly because taphonomic studies at Pirapozinho site suggested a riverine ephemerous system (*Soares et al., 1980*; *Fulfaro & Perinotto, 1996*; *Fernandes & Coimbra, 2000*; *Henriques et al., 2002*; *Henriques et al., 2005*; *Suárez, 2002*; *Bertini et al., 2006*; *Henriques, 2006*) and fossils that experienced little transportation (*Bertini et al., 2006*), thus it is more likely that *B. elegans* was a semi-aquatic turtle, similar to the extant freshwater turtles. Secondly, the general pattern observed revealed form and shape changes in both temporal and lateral emargination (upper and lower adductor chamber, respectively): as a whole, *B. elegans* skull seems to become more emarginated, flattened and longer as it grows, according to the skull shape for aquatic turtles found by *Claude et al. (2004)*, and indicating greater area

and volume for muscles attachment. In addition, the deeper temporal emargination of *B. elegans* indicates a greater increase in muscle volume (*Kischlat, 1994*), thus leading to a stronger bite force (*Sterli & De la Fuents, 2010*). This leads us to interpret such changes as related to a shift in diet as individuals grow instead of a shift in habitat.

*Malvasio et al. (2003)* described diet changes in *Podocnemis expansa*, *P. unifilis* and *P. sexturberculata* due to aging, concluding that the latter is a carnivore species, whereas the two former are omnivorous. Whereas *P. expansa* changes its diet becoming more herbivorous, *P. unifilis* remains more balanced with similar ingestion of vegetables and meat (*Malvasio et al., 2003*). *Kischlat (1994)* suggested that *B. elegans* might have fed of hard preys and, given the several mollusk and crustacean species described for the Pirapozinho site (*Dias-Brito et al., 2001*), it might have composed the diet of *B. elegans*. In this context, we agree with *Kischlat (1994)* and suggest that smaller juveniles individuals might have fed on less hard and small food items (e.g., snails and small fishes) whereas bigger old specimens fed on harder and larger preys, such as crustaceans and bigger mollusks.

Although there is a possibility that size differences could be due to sexual dimorphism as aforementioned (see Introduction, 'Geological settings and taphonomic context of the Tartaruguito site'), we were not able to assume such assumption. Furthermore, if there is size-related dimorphism, it would imply on potential diet differentiation between adults male and female of *B. elegans*. Since we were not able to determine size-related sexual dimorphism, such a statement is merely speculative.

## CONCLUSIONS

As in *Romano & Azevedo (2007)* (for shell material), our data did not show enough morphometrical variation to suggest population differences among our sample. Therefore, we did not have evidence to disprove that the "Tartaruguito" site is composed of a single population of *B. elegans*. However, it is feasible to assume that different generations of individuals were crowded in this locality by the accumulation of corpses due to several drying events as previously suggested by *Henriques et al. (2005)* and *Henriques (2006)*. Since none *B. elegans* hatchling were found in the "Tartaruguito" site until now, it might have been preferentially a freshwater foraging area.

As regards to the morphometric data, the observed variation and allometries in the skull bones, mainly the PA, QJ, SQ, QU, PO, JU, MX, PAL and PT, as well as PCAs loadings, reflect shape differences in both upper and lower adductor chambers. We interpret this allometric variation as an indicative of more area attachment and resistance for stronger adductor muscles, which are accompanied by changes in diet during aging, from softer to harder prey, as seen in living turtles species.

In regard to the use of images for carrying out morphometrics studies, we conclude that the use of calipers can be replaced by softwares that work on images. ImageJ is a useful and time-saving tool for this matter. However, one needs to beware when measuring straight lines between landmarks that are located in different depths, which result in angled lines against the projection orthogonal plane. In attention to this detail will lead to assess lower values for a given measurement than its real size.

In regard to the approaches applied to our data to deal with missing entries in the matrix (i.e., mean value or iterative imputation), both were useful for answering the questions we raised (i.e., the single population hypothesis), though little different results were obtained (few specimens out of 95% confidence ellipse in mean value approach in contrast with none specimen out of ellipse in iterative imputation approach). However, we recommend the iterative imputation as the most appropriate approach to deal with missing data in paleontological studies on the basis of the statistical assumptions it was developed (a sample-based regression for characters estimation) and the more conservative results.

**Institutional Abbreviations**

| | |
|---|---|
| **AMNH** | American Museum of Natural History, New York, NY, United States |
| **LPRP** | Laboratório de Paleontologia da Faculdade de Filosofia, Ciências e Letras de Ribeirão Preto, Universidade de São Paulo, Ribeirão Preto, SP, Brazil |
| **MN** | Museu Nacional, Universidade Federal do Rio de Janeiro, Rio de Janeiro, RJ, Brazil |
| **MCT** | Museu de Ciências da Terra, Departamento Nacional de Produção Mineral, Rio de Janeiro, RJ, Brazil |
| **MCZ** | Museum of Comparative Zoology, Harvard University, Cambridge, MA, United States |
| **MZSP** | Museu de Zoologia, Universidade de São Paulo, São Paulo, SP, Brazil. |

**Anatomical abbreviations**

| | |
|---|---|
| **PF** | prefrontal |
| **FR** | frontal |
| **PA** | parietal |
| **VO** | vomer |
| **PAL** | palatine |
| **PT** | pterygoid |
| **BS** | basisphenoid |
| **BO** | basioccipital |
| **MX** | maxilla |
| **JU** | jugal |
| **QJ** | quadratojugal |
| **QU** | quadrate |
| **PO** | postorbital |
| **SQ** | squamosal |
| **OP** | opisthotic |
| **CO** | choanal |

**Measurements abbreviations**

| | |
|---|---|
| **TLS** | Total length of skull |
| **LPF** | Length of prefrontal |
| **LFR** | Length of frontal |

| | |
|---|---|
| **LPA** | Length of parietal |
| **LVO** | Length of vomer |
| **LPAL** | Length of palatine |
| **LPT** | Length of pterygoid |
| **LBS** | Length of basisphenoid |
| **LBO** | Length of basioccipital |
| **LMX** | Length of maxilla |
| **LJU** | Length of jugal |
| **LQJ** | Length of quadratojugal |
| **LQU** | Length of quadrate |
| **LPO** | Length of postorbital |
| **LSQ** | Length of squamosal |
| **TWS** | Total width of skull |
| **WPF** | Width of prefrontal |
| **WFR** | Width of frontal |
| **WPA** | Width of parietal |
| **SMX** | Stretch of maxilla |
| **WVO** | Width of vomer |
| **WCO** | Width of choanal |
| **WPAL** | Width of palatine |
| **WBS** | Width of basisphenoid |
| **WMX** | Width of maxilla |
| **WJU** | Width of jugal |
| **WQJ** | Width of quadratojugal |
| **WPO** | Width of postorbital |
| **WOP** | Width of opisthotic. |

## ACKNOWLEDGEMENTS

We are grateful to Sergio Azevedo, Deise Henriques, Luciana Carvalho, Lílian Cruz (DGP/MN) and Max Langer (LPRP/USP) for allowing the loan of the material and/or visits to collections under their care when necessary. Pedro Romano thanks the following people and institutions for facilitating access to collections: E Gaffney, C Mehling and F Ippolito (AMNH); and R Cassab and R Machado (MCT). We thank to Gustavo Oliveira (UFRPE) for being part of Thiago Mariani's undergraduate thesis committee and for making revisions, suggestions, and comments that contributed to this paper. We are also grateful to M Lambertz (University of Bonn) and C Mariani for revisions and comments on early versions of the manuscript. Gabriel Ferreira (LPRP/USP), Mirian Menegazzo (Unesp) and Gustavo Oliveira (UFRPE) shared information about Bauru Basin fossil turtle records that were incorporated to Fig. 1. Torsten Scheyer and a second anonymous reviewer provided comments and insights that greatly improved the manuscript. Preliminary results of this paper composed the undergraduate thesis of Thiago Mariani.

### Funding

This research was supported by Conselho Nacional de Pesquisa e Desenvolvimento Tecnológico (CNPq) and Coordenação de Aperfeiçoamento de Pessoal de Nível Superior (CAPES), scholarships to Thiago Mariani. Publication fees were supported by Fundação de Amparo à Pesquisa de Minas Gerais (FAPEMIG) (PRI-00076-17 granted to Pedro Romano). The funders had no role in study design, data collection and analysis, decision to publish, or preparation of the manuscript.

### Grant Disclosures

The following grant information was disclosed by the authors:
Conselho Nacional de Pesquisa e Desenvolvimento Tecnológico (CNPq).
Coordenação de Aperfeiçoamento de Pessoal de Nível Superior (CAPES).
Fundação de Amparo à Pesquisa de Minas Gerais FAPEMIG: PRI-00076-17.

### Competing Interests

The authors declare they have no competing interests.

### Author Contributions

- Thiago F. Mariani conceived and designed the experiments, performed the experiments, analyzed the data, contributed reagents/materials/analysis tools, wrote the paper, prepared figures and/or tables, reviewed drafts of the paper.
- Pedro S.R. Romano conceived and designed the experiments, analyzed the data, contributed reagents/materials/analysis tools, wrote the paper, prepared figures and/or tables, reviewed drafts of the paper.

### Data Availability

   The raw data has been supplied as a Supplementary File.

### Supplemental Information

Supplemental information for this article can be found online at http://dx.doi.org/10.7717/peerj.2890#supplemental-information.

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
