# Peer review of "Intra-specific variation and allometry of the skull of Late Cretaceous side-necked turtle Bauruemys elegans (Pleurodira, Podocnemididae) and how to deal with morphometric data in fossil vertebrates"

_PeerJ, doi:10.7717/peerj.2890_

## Round 0.1 · original submission · Major Revisions

Dear authors,

I have received two reviews for your manuscript. Overall, both reviewers were positive about your study, its contents and perceived interest to the general audience of PeerJ. However, both reviewers have highlighted several key points that need to be improved, and I fully agree with their suggestions. Please carefully address these points in your revision: particularly, both reviewers have noted that the manuscript could be streamlined and the analyses could be organized throughout the manuscript in a more reader-friendly and straightforward manner. This is important to prevent confusion and misinterpretation of your results, and these changes will improve accessibility. In its present form, some sections of the manuscript are overlong and require shortening (please pay careful attention to the extensive descriptions regarding comparing ImageJ and caliper measurements).
Please find some additional editorial comments in the attached pdf.

I look forward to receiving a revised version of your manuscript in the near future.

·

Basic reporting

The authors provide a study of intraspecific variation of Bauruemys skulls. It is unclear to me, however, why this has to be done on the population level. In many palaeontological studies this is done on the species-level only. Similarly it is not quite clear to me if studies exist, that show actual differences in variation between the population and the species level, which could be used for comparison with the fossil data.
The title is confusing and should be changed.
The figures are appropriate in number and style and labelling.
The raw data of the measurements are provided in table form.
The literature used is up-to-date.
Several passages of the paper could be streamlined – For example the discussion why measurements on photos taken by ImageJ instead of with callipers could have been shorter as a single section in the Methods part instead of separate points in the Results and Discussion sections.
The English of the present manuscript needs to be greatly improved - I made some corrections, but not being a native speaker, I might have missed quite a bit.

Experimental design

The authors plausibly explain why they did not follow a geometric morphometric approach, but instead opt for line measurements of skull bones. To my understanding the statistical methods used (ANOVA and PCA) are appropriate for the study and the results are sound, as is the used taxonomy and the osteology.
The Objectives section should be partially rewritten so the research objectives become more clear to the reader.

Validity of the findings

To my understanding data and images provided are sound, as are the statistical analyses.

Additional comments

The largest problem I had with the present contribution is the lengthiness or wordiness of some parts. It sometimes takes the authors quite a bit to get down to the points they want to convey. I got lost several times, because the paper reads more like a thesis chapter, including an exhaustive set of citations, etc. The calliper vs. photography measurements could be shortened, similar to the discussion why it is necessary to study intraspecific variation on the population level instead of the community/species level. The last chapter 4.4 of the Discussion should be largely rewritten and shortened because it includes many repetitions.
I have made additional, specific comments directly on the PDF.

Reviewer 2 ·

Basic reporting

This is a well-written case study of 2-dimensional morphometrics in a easy (or economic) way using fossil photographs and measurements by ImageJ software. To see further comments please see attached PDF file.

Experimental design

There is still room for improvement in statistics, especially in allometric analysis. At least p-value should be given to confirm the statistic significance of regression line.

Validity of the findings

Allometric change found in this research is potentially important finding concerning ontogenetic change of feeding habit in fossil record, provided that the supporting regression lines are statistically significant. PCA analysis does not reject single-population hypothesis of Tartaruguito fossil turtle assemblage, but still there is not any strong supporting evidence, so it is not highly valuable finding.

Additional comments

Please improve readability Abstract and Material and Method section, by unifying numbering of analyses.

Annotated reviews are not available for download in order to protect the identity of reviewers who chose to remain anonymous.

---

## Round 0.2 · Minor Revisions

Dear authors,

Thank you for careful attention to the earlier comments and suggestions made by the reviewers. The re-reviewer has noted that this version is much improved, and I agree with their comments. I will be happy to accept your manuscript for publication following correction of a few minor typos (see also Reviewer #1) and editorial points, please see below:

Ln25: should be “in paleontological”
Ln26: remove colon after “e.g.”
Ln27: delete “statistic”
Ln30: I suggest to replace the first sentence with “PCA is a method to ordinate multivariate data”
Ln46: replace “portion” with “part”
Ln51: should be “most skull materials”, also please insert a space after “rare”
Ln52: please insert a space after “comparisons”
Ln60: please change to “the time”
Ln64: should be “In a”
Ln65: should be “from the”
Ln79: replace “is due to” with “reflects”
Ln81: “discovered Middle”
Ln90: “sediments that”
Ln103: “assumed that”
Ln112: use “proposed” or “suggested” rather “pointed”
Ln115: replace “bigger” with “larger”
Ln119: I suggest changing to “that sexual dimorphism may affect measurements captured in this study,…”
Ln130-131: I suggest replacing with “Efforts to study fossil materials may be hampered by difficulty in accessing foreign collections”
Ln189: delete “the” before “ImageJ”
Ln193: insert “where” after “cases”
Ln204: “or width”
Ln214: replace “its” with “their”
Ln246-7: “most variable in comparison with others” and “variation scale in these characters” – please rephrase – comparison to others is unclear here – other measurements, other species?
Ln292: insert “reflected” after “PC2”
Ln245: “The results…” should be a new paragraph below the heading
Ln271: replace “change” with “variation”
Ln272: remove extra period

·

Basic reporting

I think the manuscript has improved greatly compared to the first submission (but see list of specific comments)

The figures are relevant and informative

Experimental design

The data taken are adequate for the proposed study

Validity of the findings

the findings are robust and the interpretations are supported by the data

Additional comments

There are a few typos (and other aspects) present that should be addressed:
1) last line of chapter 1.2:
"Hennig’s (1966) semaphoront concept"

2) last paragraph in chapter 2.2.1:
"the in situ study of the material"

3) chapter 4.1.2:
"that caused it to be larger than it really was (Fig. 8A)"

4) reference Joyce and Lyson 2010:
"A neglected lineage of North American turtles"

5) Acknowledgements:
"Pedro Romano thanks the following people"

6) Acknowledgements:
"We thank Gustavo Oliveira (UFRPE) for being part of Thiago Mariani’s undergraduate thesis committee"

7) References:
Link in Rasband 1997 does not work – please try:
"Avaiable at https://imagej.nih.gov/ij/";

General Comment:
Due to the track changes funtion, some spaces are missing in the text now. Please read again carefully the word file after accepting the changes and add spaces where necessary.

---

## Round 0.3 · accepted · Accept

There are a few minor wording corrections which you can fix in production:

ln26: replace "methodologies" with "methods" (methodology is the study of methods)
ln167: replace "perform" with "collect"
ln207: replace "wide-scale" with "large"
ln246: replace "interpreted as a fail to the attempt" with "interpreted as a failure to falsify the single population hypothesis"
ln384: replace "since all analyzed skulls" with "since all skulls analyzed here"
ln451: replace "in B.elegans bigger adults" with "in larger adults of B.elegans"
ln600: replace "suffered" with "experienced"
ln601: replace "has been a" with "was a"